# Training Speedups via Batching for Geometric Learning: An Analysis of Static and Dynamic Algorithms

Daniel T. Speckhard[1,2]     Tim Bechtel[1,2]     Sebastian Kehl[3]     Jonathan Godwin[2,4]

Claudia Draxl[1,2]

[1]Humboldt-Universität zu Berlin     [2]Max Planck Institute for Solid State Research
[3]Max Planck Computing and Data Facility     [4]Orbital Materials
dts@physik.hu-berlin.edu

Reviewed on OpenReview: https://openreview.net/forum?id=v8rC6EEUep

## Abstract

Graph neural networks (GNN) have shown promising results for several domains such as materials science, chemistry, and the social sciences. GNN models often contain millions of parameters, and like other neural network (NN) models, are often fed only a fraction of the graphs that make up the training dataset in batches to update model parameters. The effect of batching algorithms on training time and model performance has been thoroughly explored for NNs but not yet for GNNs. We analyze two different batching algorithms for graph-based models, namely static and dynamic batching for two datasets, the QM9 dataset of small molecules and the AFLOW materials database. Our experiments show that changing the batching algorithm can provide up to a 2.7x speedup, but the fastest algorithm depends on the data, model, batch size, hardware, and number of training steps run. Experiments show that for a select number of combinations of batch size, dataset, and model, significant differences in model learning metrics are observed between static and dynamic batching algorithms.

## 1 Introduction

Graph neural network (GNN) models have recently shown great promise in regression and classification tasks, where the input data can be represented as graphs (21; 43). These methods have been applied to predict molecular and material behavior from nanometer to millimeter scale (37; 20; 34; 30). These tasks are of utmost importance to society, since they have opened up a research path towards exploring new molecules for drug design and carbon capture (9) and finding new materials for energy storage and generation (35).

Similar to neural network (NN) models, GNN models typically contain millions of parameters and require large training datasets to achieve sufficient predictive power. In order to effectively train GNNs on large datasets, the graph structured data needs to be batched, otherwise each update over the entire dataset takes too long for the model parameters to converge in a reasonable amount of time. In this paper, we examine two different batching techniques for graph networks, static and dynamic batching. The two algorithms differ in that static batching always grabs the same number of graphs whereas dynamic batching adds graphs to the batch conditionally and seeks to ensure that the memory occupied by the batch of graphs is constant. For static-batching, we look at three algorithms, the static-64, static-$2^N$ and static-constant algorithms.

With the advent of neural architecture searches in different fields, including with graph structured data, it is often the case that researchers train model candidate architectures several thousand times (45; 16; 40). End-users are likely to re-train the resulting model and re-tune hyper-parameters on new datasets. The total costs involved with the GNN model search and re-training pose a significant computational, financial and

environmental burden (24; 38; 31). Therefore, any savings in training time offered by the batching algorithm are significant.

We examine the effect of the algorithms on training time and metrics. For different batch sizes, we compare the computation time required to assemble the graphs in a batch. We also analyze the time to update the model parameter weights using a batch of graphs. Finally, we monitor the effect of the batching techniques on the test metrics of the models. We do this for different batch sizes, models, and datasets. Our main contributions presented in this paper are:

- We formally introduce the static and dynamic batching algorithms and several variants thereof.

- The majority of experiments show no statistically significant differences in model learning between the algorithms, except for a single combination of model, batch size and dataset.

- We find the static-constant algorithm to be the slowest algorithm and recommend not using it unless the goal is to maximize the robustness of the training pipeline. Across our experiments, we observe speedups of up to 12.5x when switching from the static-constant algorithm to another algorithm.

- Excluding the static-constant batching algorithm, we observe at most a 2.7x speedup in mean time per training step when switching from the slowest algorithm to the fastest.

- We demonstrate that the performance of the algorithm is dependent on the graph distribution in the dataset, model used, batch size, and number of training steps run.

- The static-64 and dynamic algorithms generally outperform the static-$2^N$ algorithms.

- We recommend using the dynamic algorithm, whose performance is similar to the static-64 algorithm, but is faster for fewer number of training steps.

## 2   Related Work

Several different batching methods, i.e., full-batching, mini-batching (3), stochastic learning (5), have been thoroughly analyzed for NNs in the literature (4). For instance, NN learning performance when using mini-batching has been examined as a function of the batch size. Computation time as a function of batch size has also been explored on GPUs (22). In Ref. (7), large batch sizes were found to typically slow down the convergence of the model parameters. In practice, researchers typically set the batch size as a hyperparameter that is found via cross validation (CV).

For NNs, updating the model weights can take up the bulk of training time through backpropagation (27). Typically, NNs operate on numeric data (or data that has been transformed into numeric values). When training the NN with a fixed mini-batch size of numeric data, batches have constant shape and memory requirements. Using JAX, this enables highly efficient training, since the gradient-update step can be compiled once at the start of the training. For convolutional neural networks (CNNs), this is not the case, and images of different pixel dimensions are often padded before being fed into the network (41). Similarly, for graph neural networks, the graphs in the dataset typically contain a wide variety of number of nodes and edges. To this end, batching algorithms have emerged, which pad batches of graphs to constant shapes. In this way, the gradient-update step for GNNs on batches can be compiled for execution on a hardware accelerator (e.g., a GPU). Two such methods have become popular, static batching and dynamic batching. The static batching methods always collect a fixed number of graphs while dynamic methods add graphs incrementally to a batch until some padding budget/target is reached.

That said, not all GNN models and libraries pad their data. M3GNet (8), a GNN trained for interatomic potentials (IAP) is written in TensorFlow and performs batching in a similar way to NNs. Its batching procedure collects a number of graphs corresponding to the batch size, and concatenates the atoms, bonds, states and indices into larger lists for the batch. However, it does not perform any padding after batching data. The pyTorch GNN (ptgnn) library implements a dynamic batching algorithm that also does not use padding (1). The algorithm adds graphs until either the batch has the desired batch size (i.e., number of

graphs) or some safety limit on the number of nodes in the batch has been reached. This safety limit ensures that the batch will fit into memory.

The pyTorch geometric library (15) implements a similar dynamic batching algorithm. The user specifies whether to use either (but not both) a total number of nodes as the batch cutoff/limit or total number of edges. The algorithm then adds graphs incrementally to the batch until the target number of graphs are in the batch (i.e., the target batch size) or the cutoff has been reached. It also allows the user to skip adding single graphs to the batch that would by themselves exceed the node/edge cutoff.

The Jraph library, which is built on JAX (6), implements a dynamic batching algorithm (17). Given a batch size, it performs a one-time estimate of the next largest multiple of 64 to which the sum of the nodes/edges in the batch should be padded. We can think of these estimates as the node/edge padding targets for each batch (i.e., the number of nodes/edges in a batch after padding). This estimation is done by sampling a random subset of the data. It then iteratively adds one graph at a time to the batch and stops if adding another graph would exceed the node/edge padding targets or the maximum number of graphs (i.e., the batch size) is already in the batch.

The Tensorflow GNN library (14) implements a static and dynamic batching algorithm. The static batching adds a fixed number of graphs to the batch and then pads to a constant padding value. The dynamic batching method, similar to Jraph, estimates a padding budget (they call it a size constraint) for the batch based on a random subset of data, and then adds graphs incrementally to the batch as long as they do not break this budget.

The static and dynamic algorithms have, to our knowledge, not been described in the literature, but solely within code repositories. Experiments to measure the training time as a function of the algorithm, batch size, model, or dataset are also to our knowledge not found in the literature. Our work seeks to describe the static and dynamic batching algorithms in sufficient detail and perform the aforementioned timing experiments using an implementation based on the Jraph library.

## 3 Preliminaries

**Graph representation and notation.** We consider a dataset of graphs $\mathcal{G} = \{G_1, G_2, \ldots, G_N\}$. Each graph $G_i = (\mathcal{V}_i, \mathcal{E}_i)$ consists of a set of nodes $\mathcal{V}_i$ and edges $\mathcal{E}_i$ (28; 23; 39). In sparse matrix formats the graph topology is defined by two integer arrays of length $|\mathcal{E}_i|$: the *senders* $s$ and *receivers* $r$, where an edge exists from node $s_k$ to $r_k$.

**Graph neural networks.** We evaluate three GNN models to get insight into any model dependent effects on our profiling results. Specifically, the SchNet model (37), a MPEU model (message passing with edge updates) (20; 2), and the PaiNN model (36) which uses equivariant features. The three models were chosen since the SchNet model is often used as a benchmark, the MPEU has shown better results on the AFLOW dataset in a relevant benchmark study (2), and the PaiNN model is widely used (25). These models vary in size. The SchNet model is the smallest with 84,768 trainable parameters. The next largest is the PaiNN model with 177,568 trainable parameters, and the MPEU model contains 2,553,472 trainable parameters.

We describe the three GNN models used to evaluate the batching algorithms in greater detail. GNN models take in a graph as input to the model. The nodes in the graph are typically represented by feature vectors $h_n^t$, where the subscript $n$ represents the specific node, and the superscript $t$ represents the number of times the vector has been updated. The SchNet model (37) is a graph convolutional NN. The node feature vectors are initialized with an embedding matrix and are updated with a node update equation (or convolution) that convolves the feature vector of the node, $h_i^t$, and feature vectors in neighborhood, $N_i$, of the node, $i$. It uses a convolutional kernel, $W(\vec{r_j} - \vec{r_i})$, which depends on the euclidean distance between nodes:

$$h_i^{t+1} = \sum_{j \in N_i} h_j^t \odot W^t(\vec{r_j} - \vec{r_i}). \tag{1}$$

We also look at a message passing NN model with edge updates (MPEU) (20; 2). Message passing NNs make use of a message function $M_{ij}^t$, which is defined for a sender, $i$, and receiver node, $j$, and their corresponding edge $e_{ij}^t$.

$$M_{ij}^t = f_m(h_i^t, h_j^t, e_{ij}^t) \tag{2}$$

Here, the messages are aggregated for each node with a permutation-invariant operator. The MPEU aggregates the messages with a sum,

$$m_i^{t+1} = \sum_{j \in N_i} M_{ij}^t. \tag{3}$$

In the MPEU, the edges between any two connected nodes, $i$ and $j$, are represented by edge vectors $e_{ij}^t$. The message function is a function of the edge vector,

$$M_{ij}^t = f_m^t(h_j^t, e_{ij}^t) = (W_1^t h_j^t) \odot \sigma(W_3^t \sigma(W_2^t e_{ij}^t)). \tag{4}$$

where, the feature vectors, $h_i$, are updated in the MPEU as:

$$h_n^{t+1} = S_t(h_n^t, m_n^{t+1}) = h_n^t + W_5^t \sigma(W_4^t m_n^{t+1}). \tag{5}$$

$\odot$ denotes element-wise multiplication and $\sigma$ represents the shifted soft plus function (44).

Finally, we evaluate the algorithms on the equivariant PaiNN model (36). This model differs from the MPEU and SchNet by learning not only scalar features, $s_i \in \mathbb{R}^{Fx1}$, for the representation of each atom but also equivariant features, $\vec{v}_i \in \mathbb{R}^{Fx3}$. The two representations are used to update each other. The scalar representation is updated with the following equation:

$$\Delta s_i^u = a_{ss}(s_i, \|V\vec{v}\|) + a_{sv}(s_i, \|V\vec{v}\|)\langle U\vec{v}, V\vec{v}\rangle, \tag{6}$$

where $a_{ss}$ and $a_{sv}$ represent neural networks with non-linear activation functions and $V$ and $U$ weight matrices that need to be learned. The update function for the equivariant representation is:

$$\Delta \vec{v}_i^u = a_{vv}(s_i, \|V\vec{v}\|)U\vec{v} \tag{7}$$

For all of these GNN models, the inputs is a graph. The memory requirements to store a single graph depend on the number of nodes in the graph and the number of edges. The different batching methods we will explore try to ensure that the memory requirements required to store a batch of graphs is constant, which will allow the graphs to be compiled on a GPU.

## 4 Batching Algorithms

In static batching, the method grabs a fixed number of graphs for each batch. It then checks how many nodes/edges are present in the batch. It then adds a dummy graph and adds dummy edges/nodes to this graph to serve as padding. Typically, the number of graphs included in the batch is the batch size minus one in order to leave space for the dummy graph in the batch. The algorithm pads the number of nodes/edges to try to ensure that most batches have a similar number of nodes/edges, and thereby a similar memory footprints.

**Static-constant batching.** In the static-constant algorithm, implemented in TensorFlow's GNN (TF-GNN) library, the algorithm first finds the graph with the most nodes/edges in the dataset and multiplies this number by the batch size to get a *padding target* for the nodes and edges in the batch. The padding target (or budget) is defined as $T = (T_v, T_e, T_g)$, representing the maximum allowable number of nodes, edges, and graphs in a single batch, respectively. For example, for batch size 32, the static-constant algorithm grabs 31 graphs ($T_g$), and then adds a dummy graph with enough dummy nodes/edges so that the batch has the

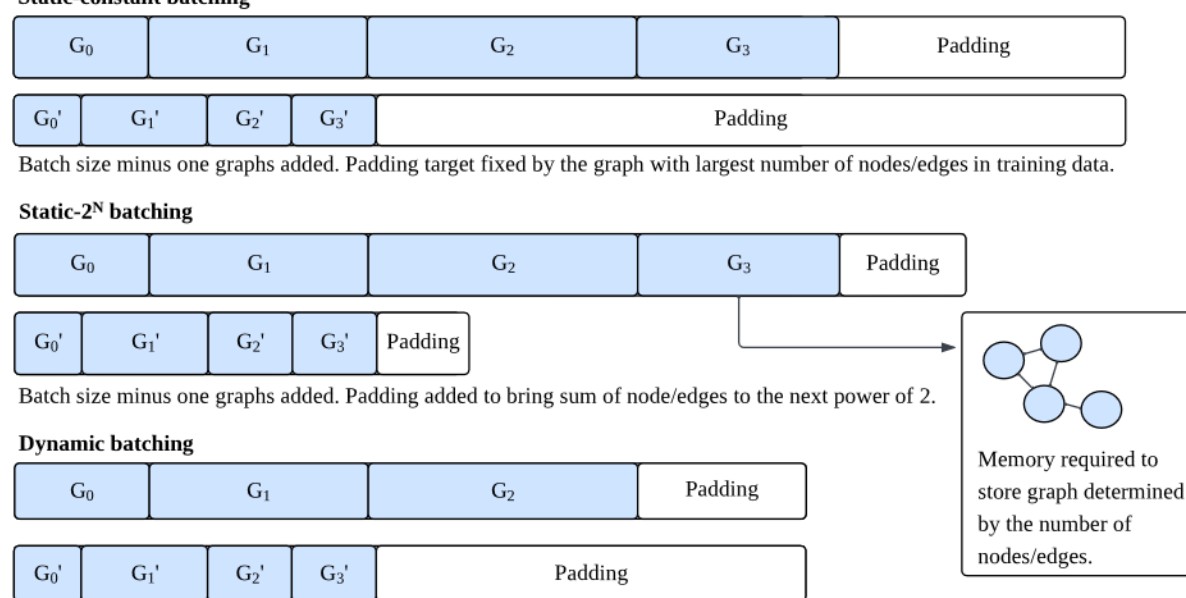

Figure 1: Visualization of the different batching algorithms for two batches each. A graph, $G_i$, is represented by a block of memory which is proportional to the number of nodes and edges in the graph. To simplify the comparison we visualize the algorithms as sampling two batches from two sets of graphs, $G$ and $G'$, that differ in the size of the graphs contained within them. Each algorithm adds padding (dummy nodes/edges and graphs) to try to keep the tensor shape of the batches constant or relatively constant. For visual clarity, the batch size for each algorithm is set to five, meaning up to four real graphs are included per batch by the algorithms. The dynamic algorithm (bottom) adds graphs to the batch iteratively only if adding the graph would not cause the number of nodes/edges/graphs to be larger than the padding target; consequently, the number of non-dummy graphs may vary. The static-$2^N$ algorithm (middle) grabs a fixed number of graphs and pads the total nodes/edges to the nearest power of two. The static-64 algorithm is not shown here for visual clarity but differs from the static-$2^N$ in that it pads the batch to the nearest multiple of 64. The static-constant algorithm (top) pads to a fixed target determined by the largest graph in the training set, often resulting in significant padding overhead.

same nodes and edges as their respective padding targets $(T_v, T_e)$. In this way, each static-constant batch occupies the same memory footprint. This means that the gradient-update step, which updates the model parameters, can always expect a batch size of a constant memory input, and can be compiled for execution on a hardware accelerator (e.g. GPU/TPU) for a fixed input tensor shape. The downside to this approach, is that for some datasets and batch sizes the target is extremely large since it is based off the largest graph in the dataset. A visualization of the algorithm can be seen in Fig. 1.

**Static-$2^N$ and static-64 algorithms.** In the static-$2^N$ algorithm, which we introduce here, the algorithm grabs batch size minus one number of graphs into a batch, and then pads the batch's total number of nodes/edges to the next largest power of two. Over the course of training, one might encounter a batch that requires a larger power of two than previously used in order to contain all of the nodes and edges in the batch. This requires the gradient-update function to recompile for a different, i.e., larger, input size. The static-$2^N$ algorithm is shown in the Appendix in Algorithm 5. Similarly, in the static-64 algorithm, which we also introduce here, the total number of nodes/edges in the batch are padded to the next largest multiple of 64.

**Padding targets estimation.** Unlike the static-constant algorithm, the padding targets in the dynamic algorithm are not based solely off the largest graph in the training dataset. The dynamic batching algorithm, in a pre-processing step, as implemented in Jraph and TF-GNN, samples a random subset of the data (e.g., 1000 graphs) to estimate padding targets for the nodes and edges. Note, this step can be done offline, and the time required to get the padding target is not included in our timing analysis of the algorithm. The padding target estimation works by calculating the mean nodes per graph $\mu_v$ and mean edges per graph $\mu_e$ from the randomly sampled subset of graphs. The edge (node) padding target $T_e$ ($T_v$) is then created by multiplying the mean number of edges (nodes) by the batch size, $N$, and then rounding to the next largest multiple of 64. The cost of this step is independent of the dataset size, however, as an upper bound, the full dataset could be used to estimate the padding target. Then, the estimation equates to a summation of node and edge counts, which is still insignificant in comparison to training time, where usually each graph is iterated over many times. The algorithm to estimate the padding target is shown in Algorithm 1, where K graphs are randomly sampled, and NextMult64 is a function to get the next multiple of 64 for an input integer. $T_g$, the graph padding target is set to the batch size minus one, which is done so that at least one dummy graph can be added later with padded nodes/edges so the batch can padded to the padding targets. The padding of the batch to the padding target is discussed in more detail in the following paragraphs.

---

**Algorithm 1** Estimate padding target (budget)

---

**Input:** Training set of graphs $\mathcal{G}$, sample size $K$, target batch size $N$
$\mathcal{S} \leftarrow \text{SampleUniform}(\mathcal{G}, K)$ {Randomly sample $K$ graphs}
$\Sigma_v \leftarrow 0, \quad \Sigma_e \leftarrow 0$
**for** $G_i \in \mathcal{S}$ **do**
    $\Sigma_v \leftarrow \Sigma_v + |V_i|$ {Accumulate node counts}
    $\Sigma_e \leftarrow \Sigma_e + |E_i|$ {Accumulate edge counts}
**end for**
$\mu_v \leftarrow \Sigma_v/K$ {Calculate mean node-count}
$\mu_e \leftarrow \Sigma_e/K$ {Calculate mean edge-count}
{Multiply means by batch size and round up to nearest 64}
$T_v \leftarrow \text{NextMult64}(\mu_v \times N)$
$T_e \leftarrow \text{NextMult64}(\mu_e \times N)$
$T_g \leftarrow N - 1$ {Reserve one slot for a padded graph}
**return** $T = (T_v, T_e, T_g)$

---

**Dynamic batching.** The dynamic algorithm, given the padding target, appends one graph at a time to the initially empty batch. Before the addition of each graph, it verifies whether a single graph contains more nodes/edges than the padding target. If so, the program terminates. The algorithm must be restarted with a larger node/edge padding target. This issue can be mitigated by looking at a larger sample of graphs when estimating the targets. Alternatively, the user can loop through the entire dataset of graphs offline and find the graph with the largest number of nodes/edges to determine suitable budgets. Provided the individual graph fits within the constraints of the padding target, the dynamic algorithm then checks if adding the graph would put the batch over the node, edge or graph padding target. If it does, the graph is not added to the batch. Instead, the algorithm adds a dummy graph with padding nodes/edges, the number of which is determined so as to pad the number of nodes/edges to the padding target. It then adds, if necessary, dummy graphs with no nodes/edges to ensure that the number of graphs in the batch is equal to the batch size. The padding function is shown in Algorithm 2. A pseudocode representation of the dynamic batching method is shown in Algorithm 3. The batch method called in the algorithm concatenates all of the nodes/edges in the list of graphs and creates a single disconnected super-graph out of the list of graphs (Pseudocode for the method is found in the Appendix, Section A).

Note that the algorithm we present here is quite different from the pyTorch-Geometric implementation, whose implementation does not perform a budget estimation (has to be entered by the user) and it does not perform a check on the number of edges, but only the nodes. We find the Jraph/TF-GNN implementation, for which a simplified version is shown in Algorithm 3, to be the more sophisticated method, and so it was used for our tests.

---

**Algorithm 2** PadToTarget

---

**Input:** List of graphs $\mathcal{B}$, Padding target $T = (T_v, T_e, T_g)$
$\Sigma_v \leftarrow 0, \quad \Sigma_e \leftarrow 0$
**for** $G_i \in \mathcal{B}$ **do**
    $\Sigma_v \leftarrow \Sigma_v + |V_i|$
    $\Sigma_e \leftarrow \Sigma_e + |E_i|$
**end for**
{Calculate deficits for nodes, edges, and graph count}
$P_v \leftarrow T_v - \Sigma_v$
$P_e \leftarrow T_e - \Sigma_e$
$P_g \leftarrow T_g - |\mathcal{B}|$
{1. Add one dummy graph containing all missing nodes/edges}
**if** $P_g > 0$ **then**
    $G_{pad} \leftarrow \text{GRAPH}(P_v, P_e)$
    $\mathcal{B} \leftarrow \mathcal{B} \cup \{G_{pad}\}$
    $P_g \leftarrow P_g - 1$
**end if**
{2. Fill remaining slots with empty graphs to reach $T_g$}
**while** $P_g > 0$ **do**
    $G_{empty} \leftarrow \text{GRAPH}(0, 0)$
    $\mathcal{B} \leftarrow \mathcal{B} \cup \{G_{empty}\}$
    $P_g \leftarrow P_g - 1$
**end while**
**return** $\mathcal{B}$

---

**Algorithm 3** Dynamic batching

---

**Input:** Training set of graphs $\mathcal{G}$, Padding Target $T = (T_v, T_e, T_g)$
$B \leftarrow \emptyset$ {Current batch of graphs}
$(c_v, c_e, c_g) \leftarrow (0, 0, 0)$ {Current counts: nodes, edges, graphs}
**for** $G_i \in \mathcal{G}$ **do**
    $(v_i, e_i) \leftarrow (|V_i|, |E_i|)$
    {1. Safety Check: Verify graph fits in budget}
    **if** $(v_i > T_v) \vee (e_i > T_e)$ **then**
        **raise** Error("Graph $G_i$ exceeds memory budget")
    **end if**
    {2. Check if adding $G_i$ overflows the current batch}
    **if** $(c_v + v_i > T_v) \vee (c_e + e_i > T_e) \vee (c_g + 1 > T_g)$ **then**
        {Yield current batch (padded) and start new one}
        **yield** $\text{BATCH}(\text{PADTOTARGET}(B, T))$
        $B \leftarrow \{G_i\}$
        $(c_v, c_e, c_g) \leftarrow (v_i, e_i, 1)$
    **else**
        {Accumulate graph into current batch}
        $B \leftarrow B \cup \{G_i\}$
        $(c_v, c_e, c_g) \leftarrow (c_v + v_i, c_e + e_i, c_g + 1)$
    **end if**
**end for**
{3. Yield any remaining graphs}
**if** $B \neq \emptyset$ **then**
    **yield** $\text{BATCH}(\text{PADTOTARGET}(B, T))$
**end if**

---

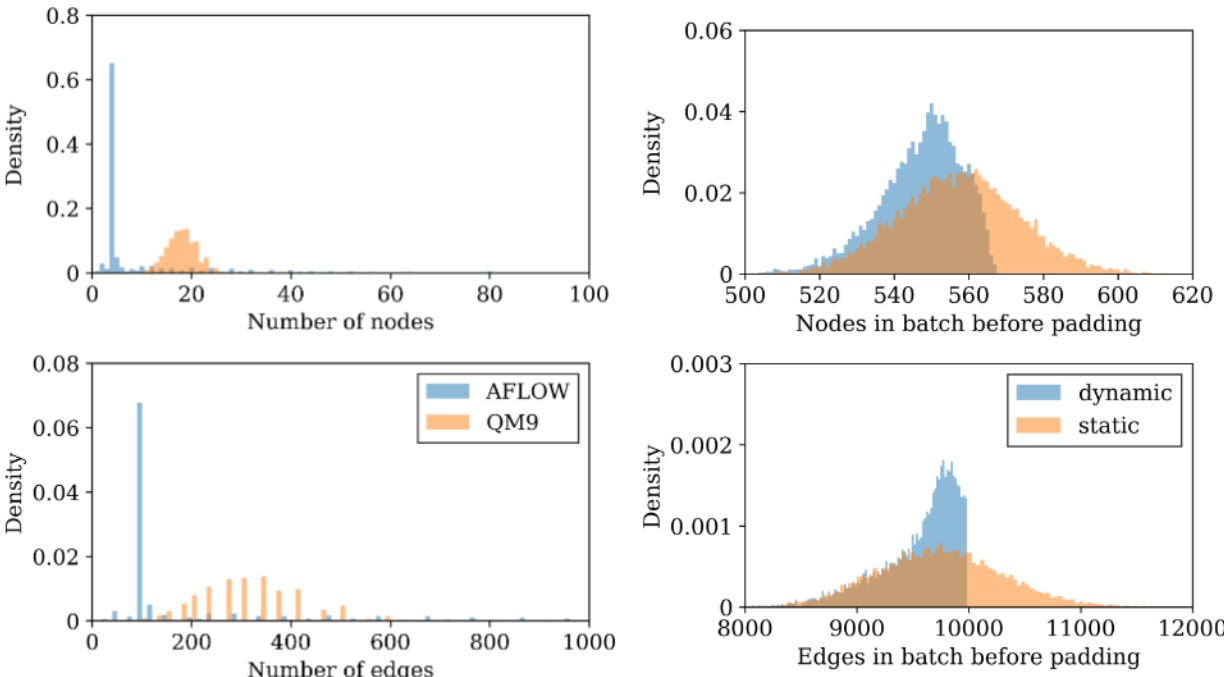

Figure 2: Left: Histograms of the number of nodes (top) and edges (bottom) in the AFLOW and QM9 datasets. Right: Histograms of the number of nodes (top) and edges (bottom) in a batch of size 32 before padding for the static-$2^N$ and dynamic batching algorithms running on the QM9 dataset.

## 5    Datasets

We run profiling experiments on two different datasets. The QM9 dataset (32) contains chemical properties (e.g., internal energies, molecular orbital energy levels) of small organic molecules. The AFLOW dataset (10) is a collection of relevant material properties (e.g., formation energies, band gaps, elastic properties). We choose these datasets for two reasons. First, they have served as benchmark datasets for GNN models (37; 26; 2). Second, they have real world applications. For instance, learning on the QM9 dataset may help models better perform targeted drug discovery, while AFLOW data may help models discover more efficient solar cell semiconductors. For QM9 we target the molecular internal energy and for AFLOW we focus on learning the enthalpy of the crystal.

The datasets are not provided in a graph format. To convert them to graphs, each atom is represented as a node in the graph. For QM9 the edges are added by fully connecting the nodes. For AFLOW, the edges are added with the K-nearest neighbor algorithm based on pairwise distance and setting K equal to 24. This KNN value was chosen from CV studies in the literature (20). From the AFLOW dataset, we remove duplicates and keep only calculations that contain both the enthalpy and the band structure, following the procedure in (2). We visualize the variety in the graph structures in the datasets in Fig. 2. From the figure we can see that the QM9 dataset node distribution appears Gaussian and centered around 17 nodes per graph. AFLOW's node distribution has long tails despite the mean of the distribution being smaller than QM9's. The distribution of edges for AFLOW and QM9 are reflective of the distribution of the nodes, this is because the number of edges are dependent on the number of nodes when using the KNN algorithm or fully connecting the graphs.

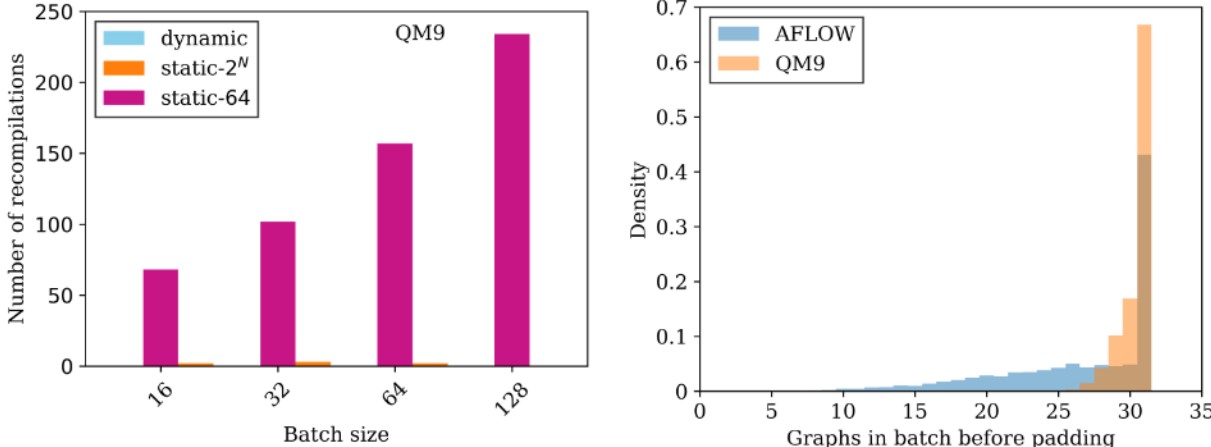

Figure 3: Left: Number of recompilations on the QM9 dataset after two million training steps in the gradient-update step as a function of batch size for different batching algorithms. Note, that these numbers are model and hardware independent. Right: Histogram of the number of graphs from the dynamic algorithm before padding in a batch size of 32 for the AFLOW and QM9 dataset.

## 6 Implementation

There are several open-source software libraries to choose from to perform these experiments. We opt to use the Jraph library (17), which is built using the JAX library (6). The JAX library traces out computations into a computation graph that it uses to optimize and compile the code. This compiled auto-differentiation code is optimized to run fast, especially on GPU. To run our experiments, we extend Jraph to run the static-$2^N$, static-64, and static-constant algorithms, log necessary profiling information and run the models we have selected for our tests.

## 7 Test setup and compute hardware

As mentioned, the benefit of dynamic batching and static-constant batching is that the memory allocated to a batch of graphs is always the same for each batch. This allows the user to compile the update functions only once and optimize the compiled binary to run on GPU. Note that with static-64 and static-$2^N$ batching, the update function can also be compiled, but needs to be recompiled for different multiples of 64 / powers of two (the static-constant algorithm does not have this problem). We time the training of models for different batching algorithms.

**Batching and weight update timing experimental setup.** We perform timing experiments on both algorithms for four batch sizes (16, 32, 64, 128) using both datasets and the three models for two million training steps. This number of training steps was chosen since it resulted in converged models (2). Experiments with fewer steps are run and the results can be found in the Appendix. We run each combination of batch size, algorithm, model, and dataset ten times to limit noise effects from the hardware in the profiling results. Each experiment reports the mean batching time, mean gradient-update step time (update time), and the sum of the two (combined time).

**Hardware used.** We run experiments on a cluster that comprises two Intel Xeon IceLake Platinum 8360Y CPUs and 4 NVIDIA A100 GPUs connected via NVLINK. We ran two sets of experiments, one using a single CPU, and the other using a single CPU and a single GPU. This design choice allows us to isolate the algorithmic overhead of the batching strategies from hardware-specific confounding variables, such as interconnect latency and multi-device synchronization costs. More discussion on porting this algorithms to a multi-GPU setup is found in the Appendix F.

# 8 Profiling results

**Batch statistics.** To understand how the batching algorithms work in practice, we analyze the statistics of the number of nodes in a batch before padding occurs. The histograms for both batching algorithms are shown for the QM9 dataset in Fig. 2 for 10,000 batches with a batch size of 32. As stated above, the dynamic batching algorithm checks the node and edge padding target, as well as the number of graphs already in the batch, before adding a graph to the batch. For this dataset and batch size, the dynamic algorithm's node padding target is 576. We can see that all dynamic batches have fewer than 576 nodes before padding. This results in a one-sided distribution, roughly a truncated Gaussian distribution, where the right-hand side is cutoff near the budget. For static batching, however, no such check exists, and we observe a roughly Gaussian distribution of nodes. Similar effects are seen in the figure for the number of edges before padding.

**Recompilation (mean vs. median).** For fewer training steps, the number of recompilations required in the gradient-update step can be the dominant factor defining the ranking of the algorithms. The number of recompilations is shown in Fig. 3 for the QM9 data for two million training steps. The number of recompilations is model and hardware independent. We see that the static-64 algorithm performs several orders of magnitude more recompilations than the static-$2^N$ and dynamic algorithms. This is to be expected, since the dynamic algorithm compiles only once, and if it encounters a graph, which is bigger than the budget, the program terminates and needs to be restarted with a larger padding target. Moreover, for static-64 batching, JAX recompiles the gradient-update step function every time a new nearest multiple of 64 is encountered when padding, and for static-$2^N$ batching every time a new nearest power of two is encountered, which is less likely for larger powers of two. For example, for the batch size of 32, the static-$2^N$ algorithm recompiles four times while the static-64 algorithm recompiles 89 times. This is expected behavior for JAX's Just-In-Time (JIT) compilation system, which caches compiled kernels for reuse but triggers a new compilation whenever it encounters a tensor shape not present in its cache (6).

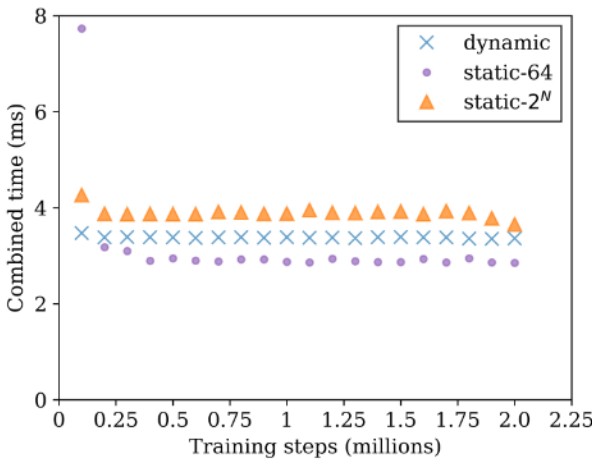

Figure 4: The running average of the combined time (sum of the batching step and gradient-update step) required per training step as a function of the total number of training steps run. Here only a single iteration is run for the batch size 32, MPEU model and QM9 dataset for the dynamic, static-64 and static-$2^N$ algorithms.

The recompilation overhead explains the discrepancy between mean and median timing results after 100,000 training steps presented in the Appendix Section C (Figs. 7 and 8). In the figures, the static-64 algorithm has the lowest median update times while the static-$2^N$ has the lowest mean update times. The mean is affected by outliers while the median is not. This tells us that when removing the effect of outliers, i.e., slow gradient-update steps which are caused by recompilations, the static-64 algorithm is fastest to update the gradients. This also tells us that the algorithm training time rankings are dependent on the dataset, where more variability in graph sizes causes more recompilations.

Fig. 4 shows the running average of the combined batching and update times per training step for QM9 data and the MPEU model with a batch size of 32. The static-64 algorithm is the slowest initially but is the fastest algorithm when training for 200,000 training steps or more. Most of the recompilations for the QM9 dataset happen in the first hundred thousand training steps. From training step one hundred thousand to two million, the static-$2^N$ algorithm does not recompile again and the static-64 algorithm recompiles only an additional twenty times. Fig. 11 in the appendix displays the same model and batch size for the AFLOW dataset and shows the dynamic (static-64) algorithm to be the fastest (slowest) initially. After 300,000 training steps the effect of the recompilations has been reduced and two algorithms are equally fast and quicker than the static-$2^N$

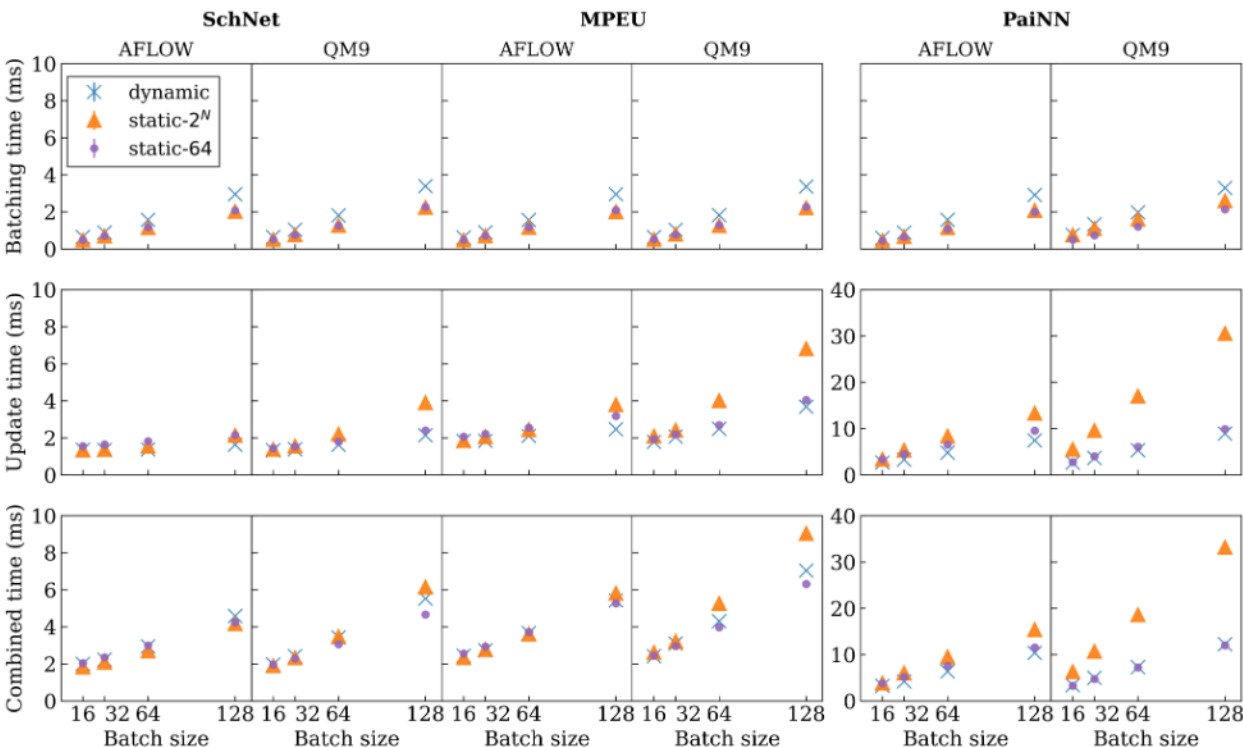

Figure 5: Timing measurements while varying the batch size for SchNet (left two columns), MPEU (middle columns) and PaiNN (right two columns). Results for the AFLOW and QM9 dataset are shown for each model. For each datapoint, ten iterations of two million training steps are run.

method. These results demonstrate that the fastest algorithm in terms of training time is dependent on the number of training steps run.

**Static-constant batching results.** The static-constant algorithm was also evaluated for a subset of data, models and batch sizes. The results are shown in the Appendix Section H. The static-constant algorithm performs poorly compared to the other static algorithms. As shown in Fig. 2, the AFLOW dataset contains a long-tailed distribution in terms of the node/edges (the estimated excess kurtosis of the two histograms are greater than one (11)). Moreover, the mean number of nodes/edges is 10-20 times smaller than the maximum number. As a result, for AFLOW, the static-constant model uses a large padding target that slows down the batching and update step. For QM9, the histograms of the dataset show that the distribution of nodes and edges resemble more closely a normal distribution. The number of edges varies from roughly one hundred to six hundred edges, and as a result, for the QM9 data, we see a smaller yet still significant slowdown using the static-constant batching algorithm compared to other methods.

**Static-64, static-$2^N$, and dynamic algorithm results.** Fig. 5 displays the breakdown of batching time per training step. We see the batching times appear roughly model and dataset independent and scale linearly with the batch size. The static algorithms loop over the number of graphs in the batch, and the dynamic algorithm does the same but may exit the loop early based on the padding target checks. Therefore it is not surprising that the batching steps scale $\mathcal{O}(N)$ where $N$ is the batch size. The dynamic batching is slower than the static-$2^N$ and static-64 algorithms due to the added overhead from the bookkeeping of the padding targets.

For the SchNet and MPEU models, the timing results of the gradient-update step at small batch sizes are similar for all algorithms. For the PaiNN model, the static-$2^N$ algorithm has a slower gradient update for all batch sizes. For larger batch sizes, across all models and datasets, the dynamic batching algorithm performs the gradient update the fastest. The dynamic batching algorithm is the fastest, since the gradient update

code is never recompiled. As seen in the histograms in Fig. 2, the number of nodes in a batch before padding is typically smaller for the dynamic algorithm. This is also true after padding. GNNs like the models we examine here, typically have a node update equation, which means that inference, and therefore the gradient-update step, scales roughly linearly with the number of nodes. When the difference between the number of nodes is small between the dynamic and static-64 algorithm we do not expect much difference in the gradient-update step time. The static-$2^N$ algorithm, however, typically adds a large number of padded nodes, which explains why the algorithm is typically slower than the dynamic algorithm in the gradient-update step. The static-64 and dynamic algorithms show roughly linear scaling in the update time with the batch size. The static-$2^N$ algorithm shows poor relative performance especially for larger batch sizes. More work needs to be done to understand the static-$2^N$ scaling behavior.

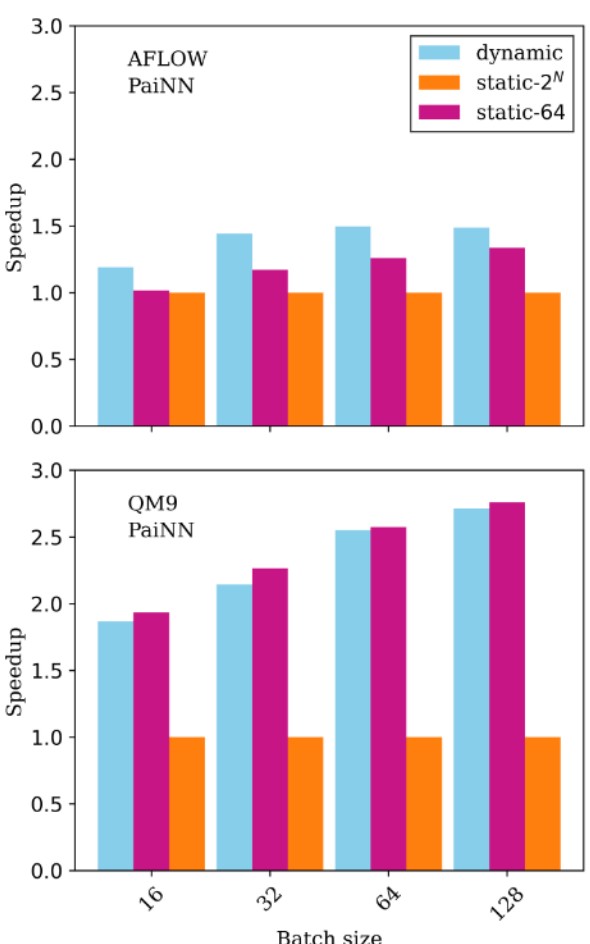

Figure 6: Speedup (GPU) when switching from the slowest algorithm in terms of combined training time per step for the PaiNN model (not including the static-constant model) for the AFLOW (top) and QM9 (bottom) datasets. For both datasets the slowest algorithm is the static-$2^N$ algorithm. If the static-constant algorithm is included the speedup increases to a maximum of 12.5x

The sum of the batching and gradient-update step mean timings, i.e., the mean combined time, gives us the mean time per training step. The speedup is defined as the ratio between the slowest algorithm combined time per step, $T_s$, and the faster algorithm combined time per step, $T_x$, as $\frac{T_s}{T_x}$ (19). For the PaiNN model, QM9 dataset, and batch size 128, the dynamic batching and static-64 algorithm have a speedup of over 2.7x when compared to the static-$2^N$ algorithm. Similarly, for the PaiNN model for AFLOW and batch sizes of 32, 64 and 128, the dynamic algorithm has a speedup of roughly 1.5x when compared to the static-$2^N$ algorithm. For the MPEU model on the QM9 dataset, we observe 1.5x speedup when using the static-64 algorithm compared to the static-$2^N$ algorithm. These observations over different datasets and models point to the large effect that changing the batching algorithm can have on the training time. The speedup from switching algorithms for the PaiNN model on the QM9 and AFLOW datasets can be seen in the Fig. 6. If we include the static-constant algorithm, which is always the slowest algorithm for the parameter space we explored here, the largest speedup is 12.5x for the PaiNN model on AFLOW with a batch size of 16 when switching to the dynamic algorithm from the static-constant algorithm.

**CPU-only performance.** We also benchmarked performance on a CPU-only setup (detailed results in Appendix Section E). Unlike the GPU setting, the gradient-update step is significantly slower on CPU and becomes the bottleneck. We find that the dynamic batching algorithm generally provides the fastest *mean* combined time because it avoids the compilation overhead, which is particularly expensive on CPU. However, similar to the GPU results, the static-64 algorithm often achieves the best *median* times once the compilation phase stabilizes.

**Effect on learning.** The batching algorithm may also affect the learning of the model. Dynamic batching does not always include the same number of graphs in a batch, since one or more dummy graphs are added when the node or edge budget is reached.

These padded graphs do not contribute to the gradient in the update step. The histogram in Fig. 3 depicts the distribution of the number of graphs before padding for the AFLOW and QM9 dataset after running 10,000 iterations with the dynamic algorithm. While most batches contain 31 graphs for a batch size of 32, some batches for the AFLOW dataset contain as few as ten graphs from the training data. This is likely due to the graph node and edge distributions (see Fig. 2). QM9 has significantly shorter tails than AFLOW, which explains why the distribution of the number of graphs before padding is more consistent for QM9.

To discern the effect of the algorithm on learning, we perform a pairwise Mann-Whitney U test (29) on the final RMSE test loss. This non-parametric test ranks samples from both distributions and uses these rankings to assess whether the underlying distributions are identical. A key advantage of the Mann-Whitney U test over the standard Student's t-test (18) is that it does not assume the data is normally distributed, an assumption that neural network training outcomes often violate (12). To account for the risk of false positives when testing across many configurations, we apply the Bonferroni correction (13) to the p-values within each model and dataset family. This correction adjusts the p-values in proportion to the number of pairwise comparisons performed. We consider results to be statistically significant only if the adjusted p-value is less than 0.05 (42).

Under this testing framework, we find that for the majority of combinations of dataset and batch size, the choice of batching algorithm has no statistically significant impact on model performance. For the QM9 dataset, we find no significant difference in test RMSE between dynamic and static batching for any model or batch size. Similarly, for the MPEU and PaiNN models on the AFLOW dataset, the results were statistically insignificant. However, a significant difference was identified for the SchNet model on the AFLOW dataset for batch size of 16 (adjusted $p \approx 0.0022$). This suggests that for heavy-tailed datasets, the dynamic algorithm's tendency to use smaller batches of training graphs can introduce gradient noise when the batch size is small. For completeness, we also provide the unadjusted Student's t-test results in Appendix Section I.

## 9 Discussion

We find the static-constant algorithm to perform the worst in terms of timing performance per training step. We do not recommend using the algorithm unless the user has a dataset where the maximum graph size is very similar to the mean graph size, and/or the priority is to have robustness in the training pipeline. We see in some cases a speedup of up to 12.5x when switching from the static-constant algorithm to the other algorithms. If we discard the static-constant algorithm, choosing one algorithm versus another can result in more than a 2.7x speedup in training time without observable differences in learning metrics. For larger hyperparameter searches or neural architecture searches (NAS), this speedup can save a very large amount of computational resources. Across the parameter space explored in this work, the static-64 and dynamic algorithm appear the fastest. In general, we would recommend the dynamic algorithm since it performs faster than the static-64 algorithm for fewer number of training steps, which can be important for NAS.

Diving deeper into the timing performance, the static algorithms perform the batching step faster. This is due to the added overhead from the bookkeeping of the padding targets in the dynamic algorithm. All of the algorithms show linear scaling in the batching time with respect to the batch size. The static-64 and dynamic algorithms show linear scaling in the gradient-update step, while the static-$2^N$ algorithm shows exponential scaling. This is due to the exponentially larger memory sizes required when using powers of two for padding. The gradient-update step is performed fastest by the dynamic algorithm. This is partially because the algorithm is never recompiled. Moreover, the static-$2^N$ algorithm typically has a larger number of nodes/edges in the batch, which slows down the gradient-update step.

The model's ability to learn is statistically indistinguishable across nearly all experimental settings. We observe a significant divergence in only one combination, the AFLOW dataset, batch size of 16 and the SchNet model. This isolated difference is likely due to the dynamic algorithm containing fewer labeled graphs per batch, particularly for long-tailed distributions in the small-batch regime.

We hope that this work serves to aide users to reduce the computational cost of training graph based models. Further work might focus on predicting the fastest algorithm as a function of the dataset, batch size, and

model. Finally, this work could in the future be extended to combine the batching algorithms with batching schemes that partition large graph networks across GPUs.

## Software and data availability

The software to perform the batching, profiling experiments, parsing of experiments, and plotting is found in `https://github.com/speckhard/gnn_batching/`. The complete set of pairwise Mann-Whitney U test and Student's t-test heatmaps can be found at `https://drive.google.com/drive/folders/1W4Fc0a7ne0z5vUDYgNc1_YuUNcjxwX3W?usp=sharing`.

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

## A  Batch and static batching algorithms

This section provides the static batching algorithm and the BATCH algorithm mentioned in Section 3. The BATCH method is shown in Algorithm 4. The pseudocode provided is a simplified version of the method found in Jraph, although the TF-GNN, ptgnn, and pyTorch libraries all perform something similar. The basic principle is to create a single disconnected super-graph that contains all of the information of the individual graphs in the batch. It initializes lists to collect information about the individual graphs. The lists collect node features, edge features, global features, connectivity indices (sender/receiver pairs) and graph metadata (node/edge counts). It then loops over the input graphs list, and concatenates the graph's node features, edge features, number of nodes/edges and sender/receiver indices into the respective batch lists. Finally, the method creates a new graph object where the constructor is fed the concatenated lists. Thereby, from a list of graphs, we have created one super-graph containing all of the information about the smaller graphs. The individual graphs can be rebuilt from the super-graph if desired. Note that the sender

---

**Algorithm 4** Batch construction (disjoint super-graph)

---

**Input:** List of graphs $\mathcal{B} = \{G_1, \ldots, G_M\}$
$\mathbf{X}_{all}, \mathbf{E}_{all}, \mathbf{u}_{all} \leftarrow \emptyset$ {Nodes, Edges, Globals}
$\mathbf{s}_{all}, \mathbf{r}_{all} \leftarrow \emptyset$ {Connectivity}
$\mathbf{n}_v, \mathbf{n}_e \leftarrow \emptyset$ {Counts}
$\delta \leftarrow 0$ {Node offset}
**for** $G_i \in \mathcal{B}$ **do**
    $\mathbf{x}_i, \mathbf{e}_i, \mathbf{u}_i \leftarrow \text{Features}(G_i)$
    $\mathbf{s}_i, \mathbf{r}_i \leftarrow \text{Topology}(G_i)$
    $v_i, e_i \leftarrow |V_i|, |E_i|$
    {Concatenate features}
    $\mathbf{X}_{all} \leftarrow \text{CONCAT}(\mathbf{X}_{all}, \mathbf{x}_i)$
    $\mathbf{E}_{all} \leftarrow \text{CONCAT}(\mathbf{E}_{all}, \mathbf{e}_i)$
    $\mathbf{u}_{all} \leftarrow \text{CONCAT}(\mathbf{u}_{all}, \mathbf{u}_i)$ {Globals (no offset needed)}
    {Shift indices to create block-diagonal adjacency}
    $\mathbf{s}_{all} \leftarrow \text{CONCAT}(\mathbf{s}_{all}, \mathbf{s}_i + \delta)$
    $\mathbf{r}_{all} \leftarrow \text{CONCAT}(\mathbf{r}_{all}, \mathbf{r}_i + \delta)$
    $\mathbf{n}_v \leftarrow \text{APPEND}(\mathbf{n}_v, v_i)$
    $\mathbf{n}_e \leftarrow \text{APPEND}(\mathbf{n}_e, e_i)$
    $\delta \leftarrow \delta + v_i$
**end for**
**return** $\text{GRAPH}(\mathbf{X}_{all}, \mathbf{E}_{all}, \mathbf{u}_{all}, \mathbf{s}_{all}, \mathbf{r}_{all}, \mathbf{n}_v, \mathbf{n}_e)$

---

and receiver node indices need to be offset by a running counter of the number of nodes already added into the batch.

The static-constant batching algorithm first finds the graph with the largest number of nodes and the graph with the largest number of edges. These values are saved and when rounded to the nearest multiple of 64 serve as targets for padding. The static batching algorithm is shown in Algorithm 5. The algorithm grabs a set of batch size minus one graphs, and then pads the set of graphs to the next multiple of 64 (static-64) or next power of two (static-$2^N$). Note that Algorithm 5 shows the static-$2^N$ implementation.

---

**Algorithm 5** Static batching

---

**Input:** Dataset $\mathcal{G}$, Start index $s$, Batch size $N$
$\mathcal{B} \leftarrow \emptyset$ {Initialize batch container}
$\Sigma_v \leftarrow 0, \quad \Sigma_e \leftarrow 0$
{Collect $N - 1$ real graphs (leaving 1 slot for padding)}
**for** $k = 0$ to $N - 2$ **do**
    $G_i \leftarrow \mathcal{G}[s + k]$
    $\mathcal{B} \leftarrow \mathcal{B} \cup \{G_i\}$
    $\Sigma_v \leftarrow \Sigma_v + |V_i|$
    $\Sigma_e \leftarrow \Sigma_e + |E_i|$
**end for**
{Pad to nearest power of 2 (or multiple of 64)}
$T_v \leftarrow \text{NEXTPOWER2}(\Sigma_v)$
$T_e \leftarrow \text{NEXTPOWER2}(\Sigma_e)$
$T_g \leftarrow N$
$T \leftarrow (T_v, T_e, T_g)$
**return** $\text{BATCH}(\text{PADTOTARGET}(\mathcal{B}, T))$

---

## B    Timing experiments setup

The timing experiments made use of python's time library. Timing statements were executed before batching, before the gradient-update step and after the gradient-update step. Block until ready commands were executed to ensure operations on the GPU had finished before the timing measurements were taken. We also experimented with placing timing measurements before the training loop and after the training loop (i.e., after 2 million steps in some cases). We then averaged this time by the number of training steps executed. We found this number to be within a standard deviation of the sum of our batching and gradient-update step timing results. This points to the fact that the library runs batch creation and update-kernel execution consecutively. It is possible to run these steps synchronously but this optimization was not done in this study. More details on the implementation can be found in the code repository.

## C    Mean versus median in timing measurements

For each timing experiment ten experiments are run and either the mean or median is taken. The median timing results for the MPEU model after running one hundred thousand steps is seen in Fig. 7. The mean results for the same experiments are shown in Fig. 8. The difference in the two figures shows the importance of the number of recompilations in the gradient-update method. This is because the mean is affected by outlier measurements, such as recompilations, while the median is not. The fact that for one hundred thousand training steps, the static-64 algorithm has the best median performance but not the best mean performance is due to the number of recompilations which is highest for this algorithm.

The mean and median timing results for two million steps are shown for the MPEU model in Fig. 9 and in Fig. 10 respectively. We see the mean static-64 gradient-update step times are shifted higher than the median results, showing that the recompilations still affect the mean results. However, the rankings of the mean combined times are the same for the median combined times across algorithms, suggesting that for longer training time the effect of recompilations is no longer as significant as we saw earlier for one hundred thousand steps.

## D    Profiling behavior as a function of the number of training steps run

The running average combined time per step is shown in Fig. 11 on the AFLOW dataset for the MPEU model using a batch size of 32. The dynamic algorithm is initially the fastest. The static-64 algorithm is initially significantly slower due to the large number of recompilations that happen in the first 100,000 training steps. After 300,000 training steps, the effect of the recompilations on the static-64 algorithm's performance is reduced and the static-64 and dynamic methods are roughly equally as fast.

## E    CPU-only timing results

This section provides supplementary details regarding the CPU only timing experiments. The algorithm performances are different when running only on CPU. The batching (inclusive of padding) performance is the same as when running on a system that has both a GPU and a CPU, which indicates that the batching is executed only on CPU. We experimented with trying to run part of the batching and padding code on GPU using JAX commands but found no speedup. The gradient-update step, however, is much slower on CPU. The mean timing results, from ten iterations of one hundred thousand training steps, are shown in Fig. 12 for the MPEU model. The median is shown in Fig. 13 for the same model. We ran one hundred thousand training steps instead of two million training steps since the computer cluster we used had a time limit of twelve hours for experiments. From the mean results, we see the dynamic batching algorithm is fastest. The median results, however, show that when the effect of recompilations are reduced, the static-64 algorithm is the fastest except for batch size 128 on the AFLOW dataset. This suggests that for longer training times the

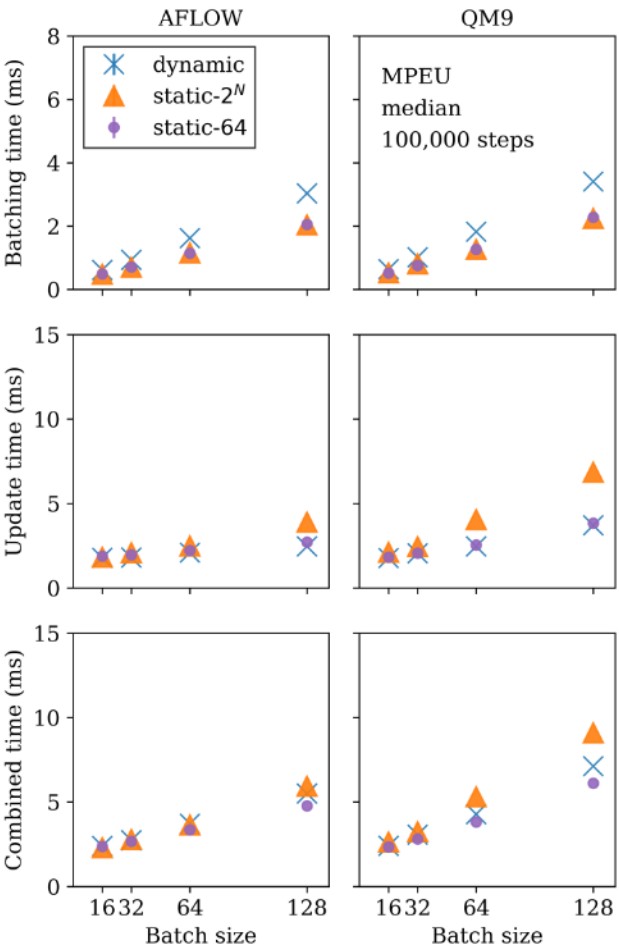

Figure 7: Median batching (top) and median gradient-update times (middle), and median combined time (bottom) for varying batch sizes on the AFLOW (left) and QM9 (right) data using the MPEU model. For each datapoint, ten iterations of one hundred thousand training steps are run.

static-64 algorithm will be the fastest. The results for the SchNet model can be seen in the accompanying code repository.

## F  Multi-GPU behavior

We utilized JAX's parallel map (`pmap`) function to assess the feasibility of the algorithms on a multi-GPU setup comprising four A100 GPUs. The dynamic batching algorithm is natively compatible with distributed setups; JAX's `pmap` distributes the graph batches across devices without requiring changes to the batching logic itself. The static-64 and static-$2^n$ algorithms, however, did not work out of the box with pmap since the algorithm expects the batches sent to each GPU to be the same size. For the AFLOW dataset, we saw that the four batches for each GPU had different shapes after padding when using the static-$2^n$ algorithm roughly 90% of the time, and close to 99% when using the static-64 algorithm. It's possible to alter the static-64 and static-$2^N$ algorithm to pad each batch of data using the largest number of nodes and edges across all batches. The results of such tests, however, will be highly implementation dependent. To simplify the comparison we evaluate the algorithms on single-GPU setups.

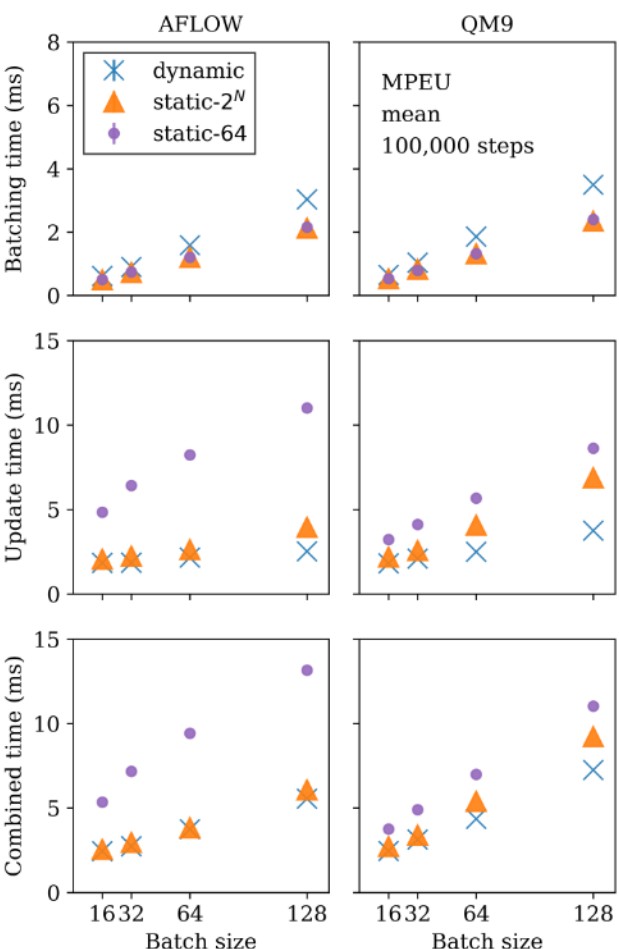

Figure 8: Mean batching (top) and mean gradient-update times (middle), and mean combined time (bottom) for varying batch sizes on the AFLOW (left) and QM9 (right) data using the MPEU model. For each datapoint, ten iterations of one hundred training steps are run.

## G   Test metric results

The mean test metric (RMSE) values as a function of the number of training steps are averaged across each of the ten iterations are averaged for each batch size, model and dataset combination and are shown in Fig. 14 for the MPEU and SchNet models using QM9 data. We do not see significant differences in the test performance for the static-$2^N$ or dynamic batching algorithm. The results for the AFLOW test dataset can be seen in Fig. 15. Note that we do not expect nor do we see any noticeable difference in learning between the static-64 and static-$2^N$ algorithms since the difference between the methods is the padding scheme, which does not affect the loss. Therefore, the static-64 RMSE curve is left out of the two figures.

## H   Static-constant batching

The TF-GNN library tutorial suggests running the static batching algorithm with a constant padding target. As described in Section 3, the algorithm first iterates through all the graphs in the dataset and saves the maximum number of nodes/edges seen in a graph and uses this as a padding target. We evaluated this method on a subset of batch sizes, dataset and model combinations, and compared it to the static-$2^N$, static-64 and dynamic algorithms. The results are shown in Table 1. Our results show that the static-

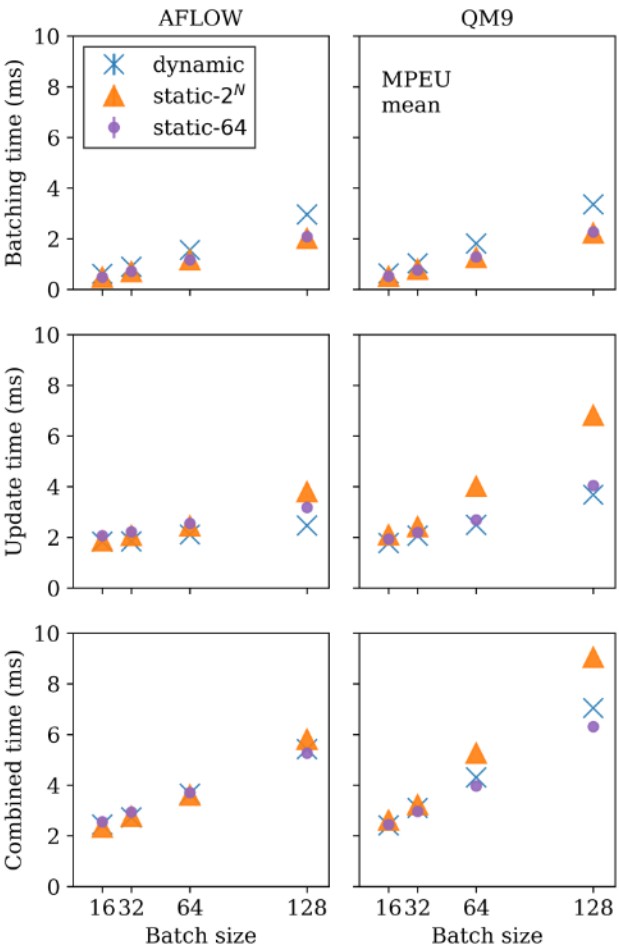

Figure 9: Mean batching time (upper row), mean gradient-update time (middle row), and the mean combined time (bottom row) for varying batch sizes on the AFLOW (left) and QM9 (right) data using the MPEU model. For each datapoint, ten iterations of two million training steps are run.

constant algorithm performs poorly in comparison to the other algorithms, and as a result it is not used in the main text.

## I   Student's t-tests on test metrics

For completeness, we also provide pairwise Student's t-test metrics on the final RMSE test loss. Note, the Student's t-test works best when the underlying distributions are normal (33), which is a strong assumption. For this reason, in the main text the Mann–Whitney U test is used and this approach is included for completeness. One can either use the t-test statistic (critical value approach) or the associated p-value to determine whether to reject the null hypothesis (18) that the test metrics after two million training steps from different batching methods belong to the same underlying distribution. We adopt p-value criteria from the literature (33), that argues that there is little evidence that the distribution of the test metrics differ significantly if the p-values are greater than 0.10. For p-values between 0.05 to 0.10, there is moderate evidence that the distributions are different. For lower p-values the evidence is strong. Ignoring the p-values that are greater than 0.10, we can look at several interesting results. For the QM9 data, SchNet model, and batch size 64, the static-$2^N$ and dynamic methods have a p-value of 0.033. For the QM9 data, SchNet model, and batch size 32, the static-$2^N$ and static-64 algorithms have p-values with the dynamic algorithm

Table 1: Static-constant timing results compared with the batching algorithms.

| Algorithm | Dataset | Model | Batch size | Batch time (ms) | Update time (ms) | Combined time (ms) |
|---|---|---|---|---|---|---|
| static-constant | AFLOW | MPEU | 16 | 0.63 | 6.34 | 7.30 |
| static-$2^N$ | AFLOW | MPEU | 16 | 0.49 | 1.89 | 2.38 |
| static-64 | AFLOW | MPEU | 16 | 0.49 | 1.89 | 2.38 |
| dynamic | AFLOW | MPEU | 16 | 0.60 | 2.41 | 3.01 |
| static-constant | AFLOW | SchNet | 32 | 0.99 | 6.34 | 7.33 |
| static-$2^N$ | AFLOW | SchNet | 32 | 0.71 | 1.38 | 2.01 |
| static-64 | AFLOW | SchNet | 32 | 0.71 | 1.64 | 2.35 |
| dynamic | AFLOW | SchNet | 32 | 0.87 | 1.30 | 2.17 |
| static-constant | AFLOW | PaiNN | 16 | 0.83 | 46.56 | 47.39 |
| static-$2^N$ | AFLOW | PaiNN | 16 | 0.45 | 3.82 | 4.27 |
| static-64 | AFLOW | PaiNN | 16 | 0.46 | 3.88 | 4.34 |
| dynamic | AFLOW | PaiNN | 16 | 0.59 | 3.20 | 3.79 |
| static-constant | QM9 | SchNet | 32 | 0.78 | 2.02 | 2.80 |
| static-$2^N$ | QM9 | SchNet | 32 | 0.74 | 1.55 | 2.29 |
| static-64 | QM9 | SchNet | 32 | 0.77 | 1.38 | 2.36 |
| dynamic | QM9 | SchNet | 32 | 1.03 | 1.38 | 2.41 |
| static-constant | QM9 | SchNet | 128 | 2.31 | 5.19 | 7.50 |
| static-$2^N$ | QM9 | SchNet | 128 | 2.20 | 3.88 | 6.08 |
| static-64 | QM9 | SchNet | 128 | 2.24 | 2.39 | 4.63 |
| dynamic | QM9 | SchNet | 128 | 3.38 | 2.13 | 5.51 |
| static-constant | QM9 | PaiNN | 16 | 0.94 | 11.67 | 12.61 |
| static-$2^N$ | QM9 | PaiNN | 16 | 0.71 | 6.43 | 7.14 |
| static-64 | QM9 | PaiNN | 16 | 0.50 | 3.30 | 3.80 |
| dynamic | QM9 | PaiNN | 16 | 0.68 | 3.33 | 4.01 |

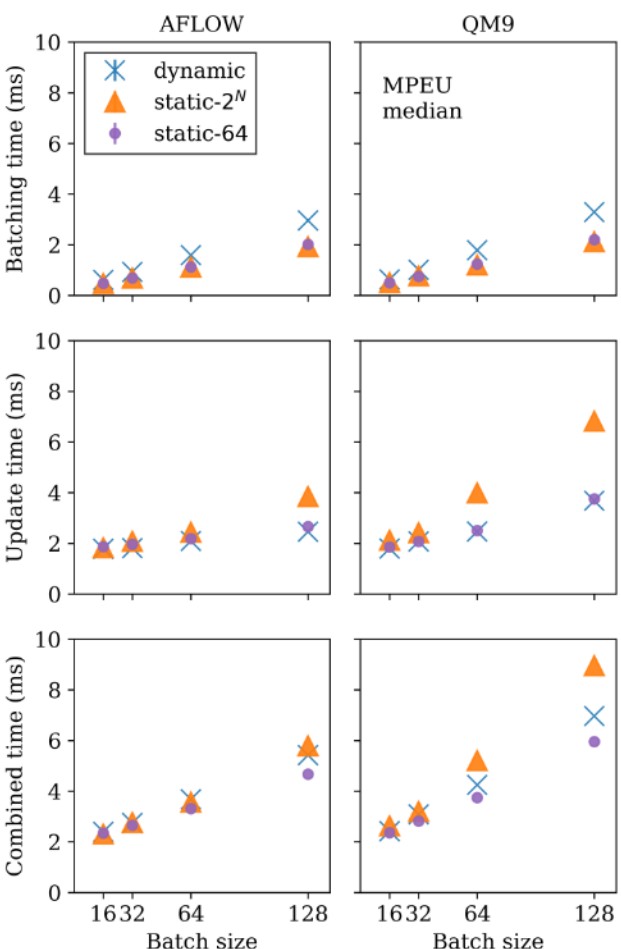

Figure 10: Median batching time (upper row), median gradient-update time (middle row), and the median combined time (bottom row) for varying batch sizes on the AFLOW (left) and QM9 (right) data using the MPEU model. For each datapoint, ten iterations of two million training steps are run.

of 0.067 and 0.053, respectively. For QM9 data, MPEU model, and batch size 64, the static-$2^N$ and dynamic methods have a p-value of 0.071. These results show that for some datasets, models, and batch sizes, the dynamic algorithm gives significantly different test metrics than the static algorithm. For the majority of combinations, however, the results are similar, which may be due to use of the Adam optimizer whose adaptive estimation of first and second moments may reduce the effect of occasionally having smaller batch sizes. Note, that unlike in the main text, a Bonferroni correction is not used in this analysis, which means we are not scaling the p-value criteria based on the number of pairwise tests that are run.

Fig. 16 shows a heatmap of pairwise t-test values and Fig. 17 shows the associated p-values across batching methods for the SchNet model, batch size 64, and QM9 dataset . Fig. 18 and Fig. 19 show a pairwise t-test heatmap of values and associated p-values respectively for the SchNet model, batch size 64 and AFLOW dataset. The complete set of plots for the other combinations of datasets, models and batch sizes can be found in the Data Availability section.

## J Mann-Whitney U-Statistic.

We show here a pairwise heatmap for the statistic values in Fig. 20 and associated p-values in Fig 21. The complete set of plots for the other combinations of datasets, models and batch sizes can be found in the Data Availability section.

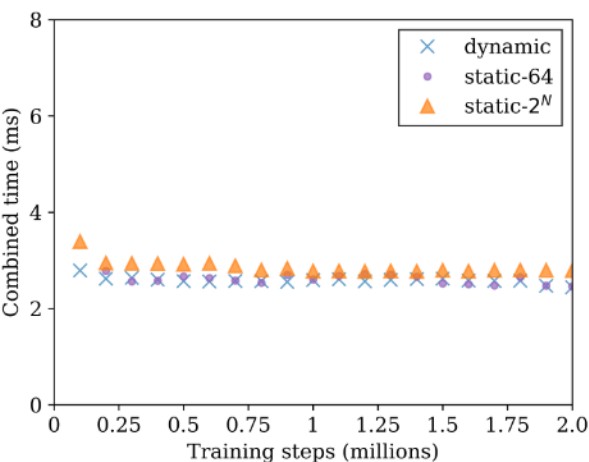

Figure 11: The running average of the combined time (sum of the batching step and gradient-update step) required per training step as a function of the total number of training steps run. Here only a single iteration is run for the batch size 32, MPEU model and AFLOW dataset for the dynamic, static-64 and static-$2^N$ algorithms.

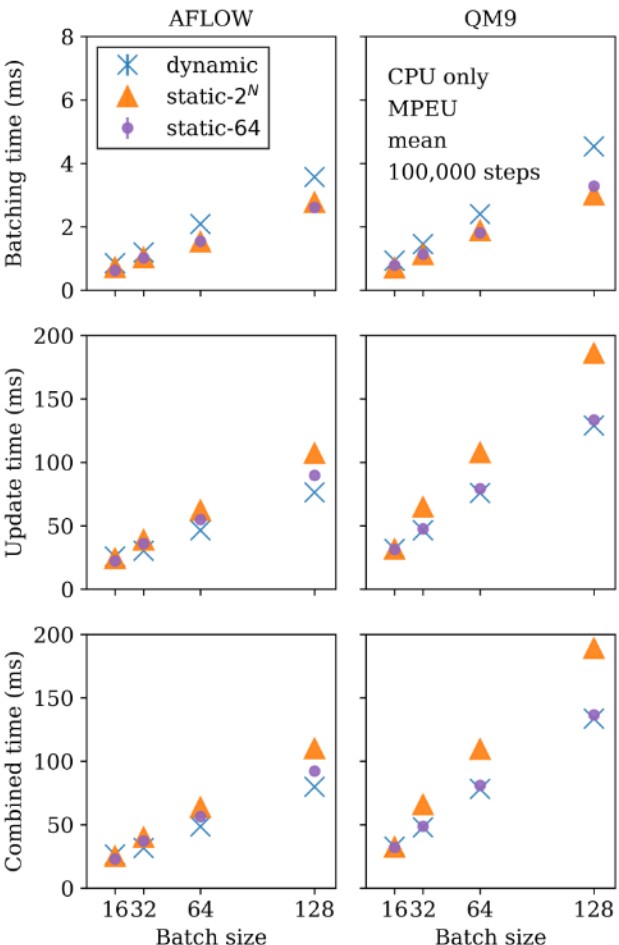

Figure 12: Mean batching time (upper row), mean gradient-update time (middle row), and the mean combined time (bottom row) on CPU for varying batch sizes on the AFLOW (left) and QM9 (right) data using the MPEU model. The experiments were run for one hundred thousand training steps.

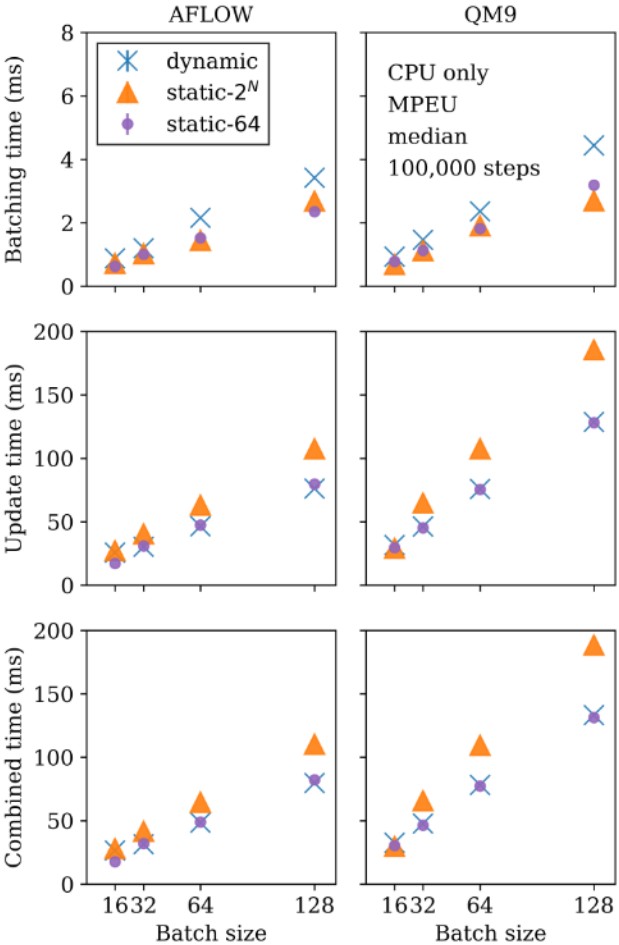

Figure 13: Median batching time (upper row), median gradient-update time (middle row), and the median combined time (bottom row) on CPU for varying batch sizes on the AFLOW (left) and QM9 (right) data using the MPEU model. The experiments were run for one hundred thousand training steps.

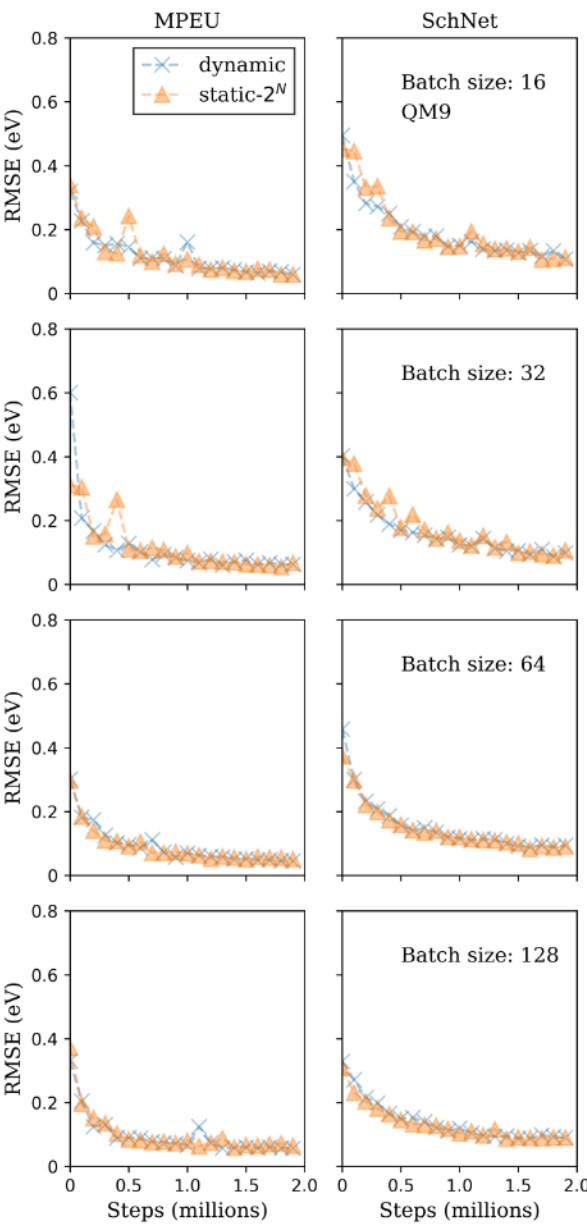

Figure 14: Mean RMSE values as a function of the number of training steps for the MPEU model (left) and SchNet (right) on QM9 test data for batch sizes of 16, 32, 64 and 128 (from top to bottom).

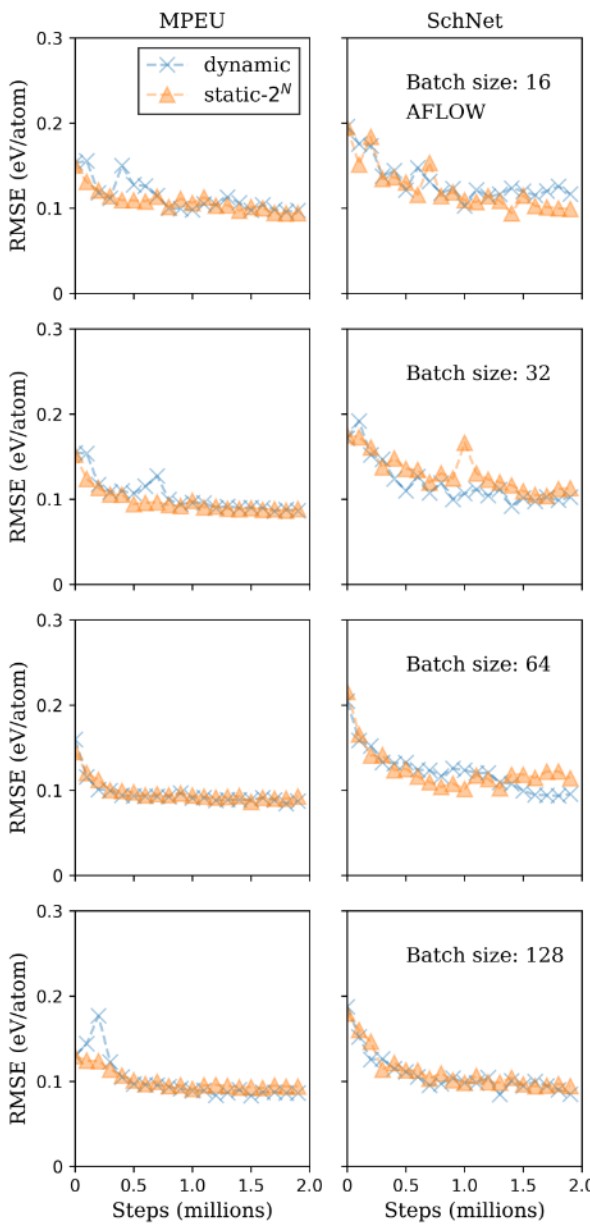

Figure 15: Mean RMSE values as a function of the number of training steps for the MPEU model (left) and SchNet (right) on AFLOW test data for batch sizes of 16, 32, 64 and 128 (from top to bottom).

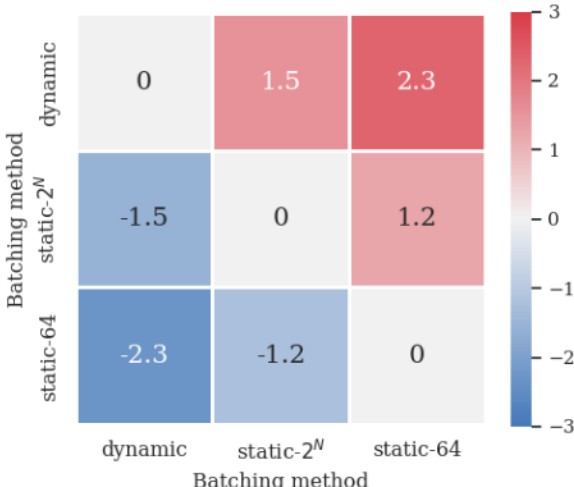

Figure 16: Heatmap of pairwise Student's t-test values on the distribution of test RMSE values for the SchNet model, QM9 data and a batch size of 64.

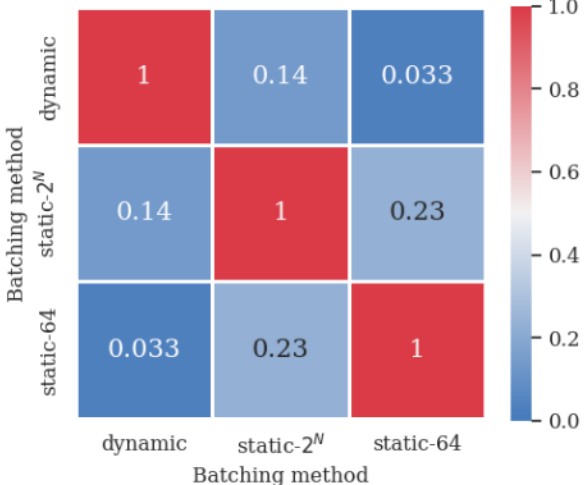

Figure 17: Heatmap of p-values from the pairwise Student's t-test on the distribution of test RMSE values for the SchNet model, QM9 data and a batch size of 64.

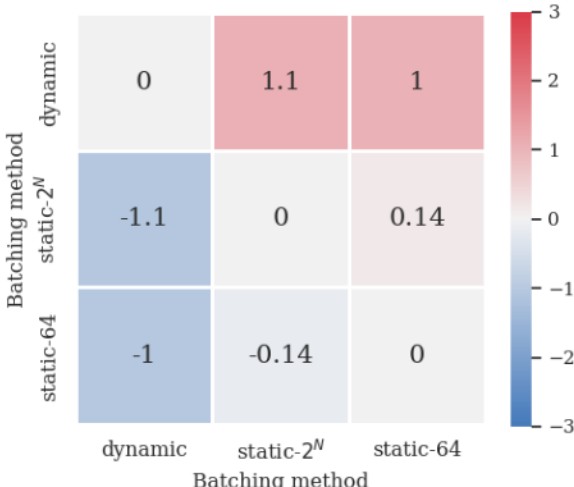

Figure 18: Heatmap of pairwise Student's t-test values on the distribution of test RMSE values for the SchNet model, AFLOW data and a batch size of 64.

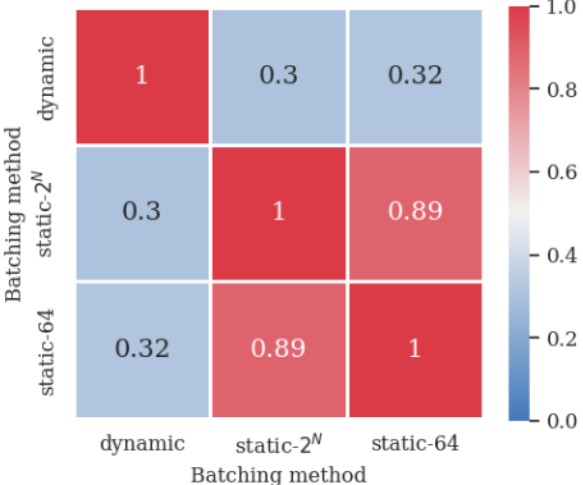

Figure 19: Heatmap of p-values from the pairwise Student's t-test on the distribution of test RMSE values for the SchNet model, AFLOW data and a batch size of 64.

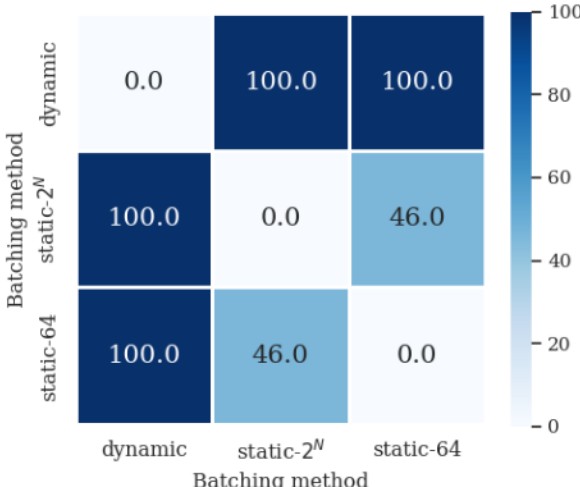

Figure 20: Heatmap of pairwise Mann-Whitney U test values on the distribution of test RMSE values for the SchNet model, AFLOW data and a batch size of 16.

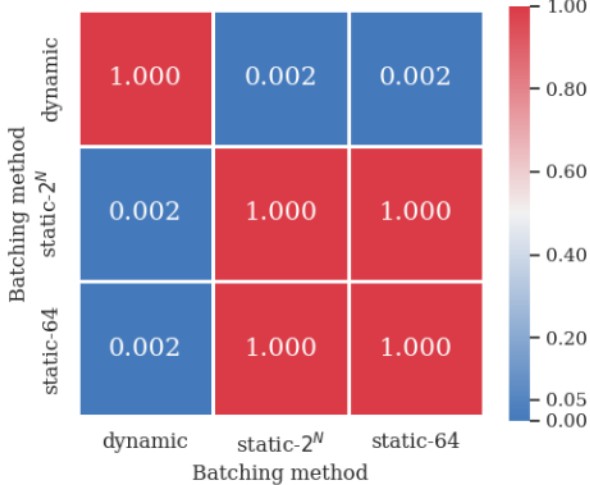

Figure 21: Heatmap of pairwise Mann-Whitney U test associated p-values with a Bonferroni correction on the distribution of test RMSE values for the SchNet model, AFLOW data and a batch size of 16.

