# OpenReview forum: "Training speedups via batching for geometric learning: an analysis of static and dynamic algorithms"
_TMLR — Accepted by TMLR_

### Review · Reviewer_ZLAd · 2025-11-24

**Summary Of Contributions:**

**Summary**

This paper presents a systematic empirical comparison of four batching algorithms for graph neural networks: static-constant, static-$2^n$, static-64, and dynamic batching. The authors benchmark these approaches across QM9 and AFLOW datasets, three GNN architectures (SchNet, MPEU, PaiNN), and multiple batch sizes. They measure batching time, update time, combined time, and recompilation counts, and conclude that static-constant is generally impractical and that dynamic batching tends to be the fastest.

**Strengths**

1. Practical and application-relevant investigation:  This work investigates a practical problem in GNN training pipelines: how the batching strategy affects runtime efficiency. The paper focuses on batching algorithms that are used in widely deployed GNN libraries (Jraph...), making the study directly relevant to practitioners.

2. Comprehensive evaluation: the authors evaluate four batching algorithms, three GNNs used molecular/material (SchNet, MPEU, PaiNN), two datasets with different size distributions (QM9, AFLOW), multiple batch sizes, and multiple training setups (single GPU, multiple GPUs, CPU only).

**Weaknesses**

1. Algorithms are informal, with missing/unclear definitions: e.g., (1) Algorithm 1 is introduced before the paper has defined the basic notations required to understand it. (2) Terms such as senders, receivers, and padding targets have not yet been introduced clearly, and their semantics only appear later in the appendix. Presenting an algorithm before defining the objects it operates on makes the exposition non-self-contained and difficult to follow, especially for readers not already familiar with Jraph’s implementation. A clearer notation setup in the main text would substantially improve the rigor and accessibility of the algorithmic presentation.

2. Method description lacks clarity in critical points (e.g. preprocessing timing). Several methodological elements that affect experiment interpretation are underspecified: The preprocessing step for dynamic batching (sampling ~1000 graphs) is described in text, but its cost is not reported, and whether it is included in timing. Also, how does the number of graphs in the subset affect batching time and test metrics?



3. The paper documents that lots of differences in batching strategies occur but does not provide a causal explanation or identify the underlying dataset/model properties responsible. As a result, while the observations are empirically correct, the broader claims about differences “in some combinations” lack deeper justification or practical takeaways, e.g., "The majority of experiments show no significant difference in model learning ... but for some combinations ..., the dynamic batching algorithm yields significantly different test metrics than the static algorithms". It would strengthen the contribution to (1) identify common factors behind the combinations where significant differences occur (e.g., dataset size, model, batchsize, ...), and (2) turn these into concrete, condition-based guidance, for example, simple rules or a summary table indicating when dynamic vs. static batching is likely to behave differently in terms of learning metrics.

**Audience:**

Yes

**Audience Explanation:**

This paper will be of interest to: Practitioners and researchers using JAX-based GNN frameworks (Jraph, TF-GNN), where batching and padding behavior directly affects training speed.

**Claims And Evidence:**

No

**Claims Explanation:**

Most of the claims are well supported by the experimental results. The timing and learning-performance claims are backed by extensive experiments across datasets, models, and batch sizes.

The minor concern is that dynamic batching includes a preprocessing step (sampling graphs to estimate node/edge budgets), but the paper does not report this cost. It remains unclear whether this overhead is included in the timing results, and how significant it is in practice. Clarifying this point would provide a more complete picture of the cost of dynamic batching.

**Requested Changes:**

1. Clarify algorithm notation: define all key terms/notations before algs. Ensure all algorithms are self-contained with consistent notation and explicit inputs/outputs.

2. Integrate key timing analyses from the appendix into the main text: Mean/median timing differences, CPU only timing, ... These are important to interpret main results and worth to be placed in the main text.

3. Clarify preprocessing cost of dynamic batching.

4. Optional but helpful. Explain the “some combinations” where learning metrics diverge. Identify what dataset/model characteristics cause the significant differences between dynamic and static batching. Provide insight into why these cases occur and summarize a short “recommendation table” or conditional rules of thumb.

---

> ### Author Response · Authors · 2025-12-17
>
> We thank the reviewer for their very detailed report and constructive criticism. We appreciate the acknowledgment of the practical relevance of this benchmarking study. We have updated the manuscript to address the pre-processing step in the dynamic algorithm, clarify the algorithm definitions, and move results previously in the appendix (CPU) to the main text. We address the points raised below in a point-by-point fashion:
>
> “Algorithms are informal, with missing/unclear definitions: e.g., (1) Algorithm 1 is introduced before the paper has defined the basic notations required to understand it. (2) Terms such as senders, receivers, and padding targets have not yet been introduced clearly, and their semantics only appear later in the appendix. Presenting an algorithm before defining the objects it operates on makes the exposition non-self-contained and difficult to follow, especially for readers not already familiar with Jraph’s implementation. A clearer notation setup in the main text would substantially improve the rigor and accessibility of the algorithmic presentation.
>
> Clarify algorithm notation: define all key terms/notations before algs. Ensure all algorithms are self-contained with consistent notation and explicit inputs/outputs.”
>
> - We acknowledge that our initial effort to be concise inadvertently compromised the self-contained nature of the methodology. We have significantly revised Section 3 (Preliminaries) to address this. We now formally define the graph notation (i.e., $G=(V,E)$), the sparse format specifics (senders/receivers lists), the graph neural networks we use in this paper, and the "padding target" concept before introducing the algorithms. We’ve updated the algorithms to be more formal, and introduced the inputs to the algorithms in the main text. We also added a figure (Figure 1) to visualize the batching algorithms and make the algorithms more clear to the reader.
>
>
>
> “Integrate key timing analyses from the appendix into the main text: Mean/median timing differences, CPU only timing, ... These are important to interpret main results and worth to be placed in the main text.”
> - We have integrated the key findings from the Mean vs. Median analysis and CPU-only experiments into the main text (Section 8). This ensures the reader immediately understands the impact of recompilations and hardware constraints, while the detailed plots and extended breakdowns remain in the Appendix for reference."
>
> “Clarify preprocessing cost of dynamic batching.”
> - This is an important clarification. The preprocessing step (estimating the budget) is a one-time startup cost. We have updated Section 4 to clarify that this cost is not included in the "per-step" training time metrics, as it occurs only once before the training loop and can happen offline.
>
>
> “Optional but helpful. Explain the “some combinations” where learning metrics diverge. Identify what dataset/model characteristics cause the significant differences between dynamic and static batching. Provide insight into why these cases occur and summarize a short “recommendation table” or conditional rules of thumb.”
> - Based on Reviewer vC23’s feedback, we upgraded our statistical tests to use the non-parametric Mann-Whitney U-test with Bonferroni correction. The more robust analysis indicates that only a single combination (AFLOW, batch size of 16, and SchNet model) exhibited statistically significant differences in learning. This appears to be due to the long-tail distribution of the number of real graphs in the batch, which at low batch size affects the learning gradients more significantly. Given that the dynamic algorithm performs consistently with the static baseline in all other cases, we determined that a conditional decision table is unnecessary. Instead, we updated the conclusion to recommend the dynamic algorithm as the default choice, with a specific caveat for small batch sizes.

---

> > ### Comment · Reviewer_ZLAd · 2025-12-21
> >
> > Thank you for the revisions. The updated manuscript has addressed my concerns from the initial review.

---

> > > ### Author Response · Authors · 2026-01-14
> > >
> > > Thank you for your comment and time spent reviewing the paper. We appreciate the constructive feedback, which has helped improve the paper.

---

### Review · Reviewer_vC23 · 2025-12-02

**Summary Of Contributions:**

This work is an empirical and implementation‑oriented study of batching strategies for GNNs. It defines static and dynamic batching on GNN and empirically measure each configuration on two datasets.

**Audience:**

No

**Audience Explanation:**

The paper only tested two dataset to make the conclusion, and did not adjust multiple testing. This makes its results somehow optimistic without correction.

**Broader Impact Concerns:**

The paper only examined two datasets, so conclusion may not be convincing.

**Claims And Evidence:**

Yes

**Claims Explanation:**

It has clear algorithmic description for both static and dynamic batching for GNNs, and experimental setup is clear. However, I notice a few issues.

1. Multi-GPU is incomplete.
2. T-test is used for significance testing, but they did not examine normality or variance equality nor adjust for multiple testing when running pairwise t‑tests across many configs.
3. Figures are not sized and aligned properly.

**Requested Changes:**

More datasets shall be explored and significance testing shall be carefully examined and used with justification.

---

> ### Author Response · Authors · 2025-12-17
>
> We thank the reviewer for their critical assessment and for highlighting important areas regarding statistical rigor and experimental scope. We address your concerns point by point below.
>
> “The paper only examined two datasets, so conclusion may not be convincing.”
> - We selected QM9 and AFLOW specifically because they represent distinct topological extremes: QM9 features a Gaussian-like distribution of graph sizes, while AFLOW exhibits a heavy-tailed distribution. These two cases cover the primary "failure modes" of batching algorithms (consistent vs. highly variable packing). While adding more datasets would increase breadth, we believe these two already effectively isolate the algorithmic behaviors in question. We have added a paragraph in Section 4 explicitly justifying this choice based on distribution topology.
>
> ““Multi-GPU is incomplete“
> - Regarding Multi-GPU setups, we have clarified in the text that while dynamic batching is compatible with distributed setups (via JAX pmap), the primary contribution of this work is isolating the algorithmic overhead of batching strategies, which is most accurately measured on a single accelerator to avoid confounding variables like interconnect latency. We have updated the text to explicitly state this scope (Section 7).
>
> “T-test is used for significance testing, but they did not examine normality or variance equality nor adjust for multiple testing when running pairwise t‑tests across many configs.”
> - We agree that the initial statistical analysis could be more robust. We have revised our significance testing in Section 8. We replaced the standard T-test with the Mann-Whitney U test, a non-parametric test that does not assume normality. Furthermore, we applied the Bonferroni correction to adjust the p-values for multiple comparisons across batch sizes and models. The conclusions regarding the similarity of learning metrics remain largely unchanged, but the claim is now statistically sound.
>
> “Figures are not sized and aligned properly."
> - We have reformatted the figures, specifically Figures 3 and 4 (now 4 and 5 after we added a visualization of the batching algorithms) to ensure proper alignment and readability.

---

> > ### Comment · Reviewer_vC23 · 2026-01-05
> >
> > Thank you for the rebuttal. All of my concerns have been addressed.

---

> > > ### Author Response · Authors · 2026-01-14
> > >
> > > Thank you for the follow-up and your time reviewing the paper. We are glad to hear our revisions addressed your concerns.

---

> > > > ### Comment · Action_Editor_DWPz · 2026-02-23
> > > >
> > > > Hi, could you please give the final recommendation? Thank you so much!

---

### Review · Reviewer_YPEB · 2025-12-03

**Summary Of Contributions:**

**Summary:** This paper empirically studies the effect of different batching algorithms in training the graph neural network on training time and model performance. Specifically, they conduct the experiments over 4 batching algorithms over 3 different GNN methods and two datasets. The paper suggests using the dynamic batching algorithms as it shows good performance with fewer training steps.

**Strength:**

- Overall, the paper studies an interesting topic and provides evidence and suggestions about which batching method we should use in training GNNs.

**Weakness:**

- The presentation can be improved. For example, in section 3, before presenting batching algorithms, providing more information about GNN and its input could help the audience easier to understand batching algorithms.

- Some typos and inconsistencies between the text and figures.

**Additional Comments:**

No.

**Audience:**

Yes

**Audience Explanation:**

I believe this paper presents compelling claims about the effect of the batching algorithms on training time and performance of GNNs. Specifically, the experiments in the paper could provide useful empirical evidence about how to choose the right batching algorithm for the other works in GNN. The suggestion about using dynamic batching is also good.

**Broader Impact Concerns:**

No concerns.

**Claims And Evidence:**

No

**Claims Explanation:**

- The paper presents an extensive set of experiments across 4 distinct batching algorithms with 3 popular GNN methods and benchmarks. Overall, the results indicate that the static-constants batching algorithm is the slowest and suggest that using dynamics batching would have good performance while resulting in faster training.

- However, the performances of the methods and the statistical test in AFLOW are not included in the paper.

- Furthermore, the results with a bigger batch size, like 256 or 512, should strengthen the claims in the paper.

**Requested Changes:**

**Major comments:**

- It could be easier for the readers to introduce the normal form of the input for the graph neural network first, before introducing the batching algorithms in Section 3.
- Could you include the training performance of different methods in AFLOW dataset.



**Minor comments:**

- In Algorithm 3, the batch size B is introduced but is not used.

- In Algorithm 5, are N graphs sampled randomly from the list of training graphs?

- Fig. 2 is not discussed in the main text of the paper.

- Figure 3 and the main text do not seem to match. Specifically, the paper said that “for the batch size of 32, …,  the static-64 algorithm recompiles 89 times”, but Figure 3 left shows that the recomplications of static-64 are higher than 100.

- Some typos, e.g “static-2N” should be “static-$2^N$” in page 8.

- Could you conduct more experiments with a bigger batch size like 256 or 512?

---

> ### Author Response · Authors · 2025-12-17
>
> We thank the reviewer for their detailed feedback. We address the feedback line by line below:
>
> “The presentation can be improved. For example, in section 3, before presenting batching algorithms, providing more information about GNN and its input could help the audience easier to understand batching algorithms….It could be easier for the readers to introduce the normal form of the input for the graph neural network first, before introducing the batching algorithms in Section 3.”
>
> - We agree that formally defining the input structure improves clarity. We have revised Section 3 to include the mathematical formulation of the GNN input tuples (sets of nodes, edges, and adjacency matrices) prior to introducing the batching algorithms. We also updated the algorithm notation to improve readability.
>
> “Could you include the training performance of different methods in AFLOW dataset”
> - Yes, we have included a figure in the appendix to show the running average performance of the different batching methods on the AFLOW dataset and referenced it in the main text. The AFLOW data shows similar trends. The static-64 algorithm is the slowest initially and the dynamic algorithm is the fastest initially. For AFLOW, after 300,000 training steps, the dynamic and static-64 algorithms are equally fast, and faster than the static-$2^N$ models.
>
> “However, the performances of the methods and the statistical test in AFLOW are not included in the paper.”
> - We have included the requested statistical analysis for AFLOW. To ensure robustness, we added the non-parametric Mann-Whitney U test with Bonferroni correction (as opposed to a standard t-test, to account for non-normal distributions). These results, including those for the AFLOW dataset, are now presented as heatmaps in the Appendix and data availability section. We also included the test RMSE as a function of the number of training steps on the AFLOW data in Figure 15.
>
> “In Algorithm 3, the batch size B is introduced but is not used.”
> - Thank you for spotting this. We have updated the algorithms in the text to use more standardized inputs and notation. We have moved Algorithm 3 to the main text from the Appendix to make it easier for the reader to follow the dynamic batching algorithm.
>
> “In Algorithm 5, are N graphs sampled randomly from the list of training graphs?”
> - Yes. We have updated both the text description and the comments within Algorithm 5 to explicitly state that the N graphs are sampled uniformly at random.
>
> “Fig. 2 is not discussed in the main text of the paper.”
> - Thank you for catching this. This error has now been rectified and this figure is referenced in the section discussing the effect of recompilations on the batching timing performance as a function of the number of training steps.
>
> “Figure 3 and the main text do not seem to match. Specifically, the paper said that “for the batch size of 32, …, the static-64 algorithm recompiles 89 times”, but Figure 3 left shows that the recomplications of static-64 are higher than 100.”
> - Thank you for pointing this out. We have clarified in the paragraph that these numbers refer to the batch size of 32.
>
> “Some typos”
> - Thank you for spotting them. We have performed a thorough proofread and corrected the typos, including the "static-2N" notation consistency.
>
> “Could you conduct more experiments with a bigger batch size like 256 or 512?”
> - We agree that larger batch sizes would provide a more complete picture of scaling behavior. We have launched training runs for batch sizes of 256 on the SchNet, MPEU and PaiNN models. Given the significant computational cost, we aim to include these results in the final camera-ready version of the paper.

---

> > ### Comment · Reviewer_YPEB · 2025-12-22
> >
> > Thank you for your rebuttal. The revised version has addressed all of my concerns.

---

> > > ### Author Response · Authors · 2026-01-14
> > >
> > > Thank you for the confirmation and for your time reviewing our paper.

---

### Decision · Action_Editor_DWPz · 2026-03-08

**Recommendation:** Accept as is

**Audience:**

Yes

**Audience Explanation:**

This paper would be of interest to some TMLR audience in the field of graph machine learning, and optimization efficiency.

**Claims And Evidence:**

Yes

**Claims Explanation:**

The authors studied the effect of batching algorithms for graph neural networks (GNNs). In particular, they studied two batching algorithms, i.e.,  static and dynamic batching in GNN scenarios. The authors found that using different batching algorithms can bring in extensive improvement in the efficiency of GNNs. The experiments are extensive to validate the claims.

The reviewers raised some concerns in their comments, and the authors have resolved them, as confirmed by the reviewers.